# Minimizing Regret on Reflexive Banach Spaces and Nash Equilibria in Continuous Zero-Sum Games

**Maximilian Balandat,  Walid Krichene,  Claire Tomlin,  Alexandre Bayen**
Electrical Engineering and Computer Sciences, UC Berkeley
[balandat,walid,tomlin]@eecs.berkeley.edu, bayen@berkeley.edu

## Abstract

We study a general adversarial online learning problem, in which we are given a decision set $\mathcal{X}$ in a reflexive Banach space $X$ and a sequence of reward vectors in the dual space of $X$. At each iteration, we choose an action from $\mathcal{X}$, based on the observed sequence of previous rewards. Our goal is to minimize regret. Using results from infinite dimensional convex analysis, we generalize the method of Dual Averaging to our setting and obtain upper bounds on the worst-case regret that generalize many previous results. Under the assumption of uniformly continuous rewards, we obtain explicit regret bounds in a setting where the decision set is the set of probability distributions on a compact metric space $S$. Importantly, we make no convexity assumptions on either $S$ or the reward functions. We also prove a general lower bound on the worst-case regret for any online algorithm. We then apply these results to the problem of learning in repeated two-player zero-sum games on compact metric spaces. In doing so, we first prove that if both players play a Hannan-consistent strategy, then with probability 1 the empirical distributions of play weakly converge to the set of Nash equilibria of the game. We then show that, under mild assumptions, Dual Averaging on the (infinite-dimensional) space of probability distributions indeed achieves Hannan-consistency.

## 1   Introduction

Regret analysis is a general technique for designing and analyzing algorithms for sequential decision problems in adversarial or stochastic settings (Shalev-Shwartz, 2012; Bubeck and Cesa-Bianchi, 2012). Online learning algorithms have applications in machine learning (Xiao, 2010), portfolio optimization (Cover, 1991), online convex optimization (Hazan et al., 2007) and other areas. Regret analysis also plays an important role in the study of repeated play of finite games (Hart and Mas-Colell, 2001). It is well known, for example, that in a two-player zero-sum finite game, if both players play according to a Hannan-consistent strategy (Hannan, 1957), their (marginal) empirical distributions of play almost surely converge to the set of Nash equilibria of the game (Cesa-Bianchi and Lugosi, 2006). Moreover, it can be shown that playing a strategy that achieves sublinear regret almost surely guarantees Hannan-consistency.

A natural question then is whether a similar result holds for games with infinite action sets. In this article we provide a positive answer. In particular, we prove that in a continuous two-player zero sum game over compact (not necessarily convex) metric spaces, if both players follow a Hannan-consistent strategy, then with probability 1 their empirical distributions of play weakly converge to the set of Nash equilibria of the game. This in turn raises another important question: Do algorithms that ensure Hannan-consistency exist in such a setting? More generally, can one develop algorithms that guarantee sub-linear growth of the worst-case regret? We answer these questions affirmatively as well. To this end, we develop a general framework to study the Dual Averaging (or Follow the Regularized Leader) method on reflexive Banach spaces. This framework generalizes a wide range of existing

results in the literature, including algorithms for online learning on finite sets (Arora et al., 2012) and finite-dimensional online convex optimization (Hazan et al., 2007).

Given a convex subset $\mathcal{X}$ of a reflexive Banach space $X$, the generalized Dual Averaging (DA) method maximizes, at each iteration, the cumulative past rewards (which are elements of $X^*$, the dual space of $X$) minus a regularization term $h$. We show that under certain conditions, the maximizer in the DA update is the Fréchet gradient $Dh^*$ of the regularizer's conjugate function. In doing so, we develop a novel characterization of the duality between essential strong convexity of $h$ and essential Fréchet differentiability of $h^*$ in reflexive Banach spaces, which is of independent interest.

We apply these general results to the problem of minimizing regret when the rewards are uniformly continuous functions over a compact metric space $S$. Importantly, we do not assume convexity of either $S$ or the rewards, and show that it is possible to achieve sublinear regret under a mild geometric condition on $S$ (namely, the existence of a locally $Q$-regular Borel measure). We provide explicit bounds for a class of regularizers, which guarantee sublinear worst-case regret. We also prove a general lower bound on the regret for any online algorithm and show that DA asymptotically achieves this bound up to a $\sqrt{\log t}$ factor.

Our results are related to work by Lehrer (2003) and Sridharan and Tewari (2010); Srebro et al. (2011). Lehrer (2003) gives necessary geometric conditions for Blackwell approachability in infinite-dimensional spaces, but no implementable algorithm guaranteeing Hannan-consistency. Sridharan and Tewari (2010) derive general regret bounds for Mirror Descent (MD) under the assumption that the strategy set is uniformly bounded in the norm of the Banach space. We do not make such an assumption here. In fact, this assumption does not hold in general for our applications in Section 3.

The paper is organized as follows: In Section 2 we introduce and provide a general analysis of Dual Averaging in reflexive Banach spaces. In Section 3 we apply these results to obtain explicit regret bounds on compact metric spaces with uniformly continuous reward functions. We use these results in Section 4 in the context of learning Nash equilibria in continuous two-player zero sum games, and provide a numerical example in Section 4. All proofs are given in the supplementary material.

## 2 Regret Minimization on Reflexive Banach Spaces

Consider a sequential decision problem in which we are to choose a sequence $(x_1, x_2, \dots)$ of actions from some feasible subset $\mathcal{X}$ of a reflexive Banach space $X$, and seek to maximize a sequence $(u_1(x_1), u_2(x_2), \dots)$ of *rewards*, where the $u_\tau : X \to \mathbb{R}$ are elements of a given subset $\mathcal{U} \subset X^*$, with $X^*$ the dual space of $X$. We assume that $x_t$, the action chosen at time $t$, may only depend on the sequence of previously observed reward vectors $(u_1, \dots, u_{t-1})$. We call any such algorithm an *online algorithm*. We consider the *adversarial* setting, i.e., we do not make any distributional assumptions on the rewards. In particular, they could be picked maliciously by some adversary.

The notion of *regret* is a standard measure of performance for such a sequential decision problem. For a sequence $(u_1, \dots, u_t)$ of reward vectors, and a sequence of decisions $(x_1, \dots, x_t)$ produced by an algorithm, the regret of the algorithm w.r.t. a (fixed) decision $x \in \mathcal{X}$ is the gap between the realized reward and the reward under $x$, i.e., $R_t(x) := \sum_{\tau=1}^{t} u_\tau(x) - \sum_{\tau=1}^{t} u_\tau(x_\tau)$. The *regret* is defined as $\mathcal{R}_t := \sup_{x \in \mathcal{X}} R_t(x)$. An algorithm is said to have *sublinear regret* if for any sequence $(u_t)_{t \geq 1}$ in the set of admissible reward functions $\mathcal{U}$, the regret grows sublinearly, i.e. $\limsup_t \mathcal{R}_t / t \leq 0$.

**Example 1.** *Consider a finite action set $S = \{1, \dots, n\}$, let $X = X^* = \mathbb{R}^n$, and let $\mathcal{X} = \Delta_{n-1}$, the probability simplex in $\mathbb{R}^n$. A reward function can be identified with a vector $u \in \mathbb{R}^n$, such that the $i$-th element $u_i$ is the reward of action $i$. A choice $x \in \mathcal{X}$ corresponds to a randomization over the $n$ actions in $S$. This is the classic setting of many regret-minimizing algorithms in the literature.*

**Example 2.** *Suppose $S$ is a compact metric space with $\mu$ a finite measure on $S$. Consider $X = X^* = L^2(S, \mu)$ and let $\mathcal{X} = \{x \in X : x \geq 0 \text{ a.e.}, \|x\|_1 = 1\}$. A reward function is an $L^2$-integrable function on $S$, and each choice $x \in \mathcal{X}$ corresponds to a probability distribution (absolutely continuous w.r.t. $\mu$) over $S$. We will explore a more general variant of this problem in Section 3.*

In this Section, we prove a general bound on the worst-case regret for DA. DA was introduced by Nesterov (2009) for (finite dimensional) convex optimization, and has also been applied to online learning, e.g. by Xiao (2010). In the finite dimensional case, the method solves, at each iteration, the optimization problem $x_{t+1} = \arg\max_{x \in \mathcal{X}} \left\langle \eta_t \sum_{\tau=1}^{t} u_\tau, \ x \right\rangle - h(x)$, where $h$ is a strongly convex

regularizer defined on $\mathcal{X} \subset \mathbb{R}^n$ and $(\eta_t)_{t \geq 0}$ is a sequence of learning rates. The regret analysis of the method relies on the duality between strong convexity and smoothness (Nesterov, 2009, Lemma 1). In order to generalize DA to our Banach space setting, we develop an analogous duality result in Theorem 1. In particular, we show that the correct notion of strong convexity is (uniform) essential strong convexity. Equipped with this duality result, we analyze the regret of the Dual Averaging method and derive a general bound in Theorem 2.

## 2.1 Preliminaries

Let $(X, \|\cdot\|)$ be a reflexive Banach space, and denote by $\langle \cdot, \cdot \rangle : X \times X^* \to \mathbb{R}$ the canonical pairing between $X$ and its dual space $X^*$, so that $\langle x, \xi \rangle := \xi(x)$ for all $x \in X, \xi \in X^*$. By the *effective domain* of an extended real-valued function $f : X \to [-\infty, +\infty]$ we mean the set $\operatorname{dom} f = \{x \in X : f(x) < +\infty\}$. A function $f$ is *proper* if $f > -\infty$ and $\operatorname{dom} f$ is non-empty. The *conjugate* or *Legendre-Fenchel transform* of $f$ is the function $f^* : X^* \to [-\infty, +\infty]$ given by

$$f^*(\xi) = \sup_{x \in X} \langle x, \xi \rangle - f(x) \tag{1}$$

for all $\xi \in X^*$. If $f$ is proper, lower semicontinuous and convex, its *subdifferential* $\partial f$ is the set-valued mapping $\partial f(x) = \{\xi \in X^* : f(y) \geq f(x) + \langle y - x, \xi \rangle \text{ for all } y \in X\}$. We define $\operatorname{dom} \partial f := \{x \in X : \partial f(x) \neq \emptyset\}$. Let $\Gamma$ denote the set of all convex, lower semicontinuous functions $\gamma : [0, \infty) \to [0, \infty]$ such that $\gamma(0) = 0$, and let

$$\Gamma_U := \{\gamma \in \Gamma : \forall r > 0, \ \gamma(r) > 0\} \qquad \Gamma_L := \{\gamma \in \Gamma : \gamma(r)/r \to 0, \text{ as } r \to 0\} \tag{2}$$

We now introduce some definitions. Additional results are reviewed in the supplementary material.

**Definition 1** (Strömberg, 2011). *A proper convex lower semicontinuous function $f : X \to (-\infty, \infty]$ is* essentially strongly convex *if*

  (i) *$f$ is strictly convex on every convex subset of $\operatorname{dom} \partial f$*

  (ii) *$(\partial f)^{-1}$ is locally bounded on its domain*

  (iii) *for every $x_0 \in \operatorname{dom} \partial f$ there exists $\xi_0 \in X^*$ and $\gamma \in \Gamma_U$ such that*

$$f(x) \geq f(x_0) + \langle x - x_0, \xi_0 \rangle + \gamma(\|x - x_0\|), \quad \forall x \in X. \tag{3}$$

*If (3) holds with $\gamma$ independent of $x_0$, $f$ is* uniformly essentially strongly convex *with modulus $\gamma$.*

**Definition 2** (Strömberg, 2011). *A proper convex lower semicontinuous function $f : X \to (-\infty, \infty]$ is* essentially Fréchet differentiable *if* $\operatorname{int} \operatorname{dom} f \neq \emptyset$, $f$ *is Fréchet differentiable on* $\operatorname{int} \operatorname{dom} f$ *with Fréchet derivative $Df$, and $\|Df(x_j)\|_* \to \infty$ for any sequence $(x_j)_j$ in* $\operatorname{int} \operatorname{dom} f$ *converging to some boundary point of* $\operatorname{dom} f$.

**Definition 3.** *A proper Fréchet differentiable function $f : X \to (-\infty, \infty]$ is* essentially strongly smooth *if $\forall x_0 \in \operatorname{dom} \partial f, \ \exists \xi_0 \in X^*, \ \kappa \in \Gamma_L$ such that*

$$f(x) \leq f(x_0) + \langle \xi_0, x - x_0 \rangle + \kappa(\|x - x_0\|), \quad \forall x \in X. \tag{4}$$

*If (4) holds with $\kappa$ independent of $x_0$, $f$ is* uniformly essentially strongly smooth *with modulus $\kappa$.*

With this we are now ready to give our main duality result:

**Theorem 1.** *Let $f : X \to (-\infty, +\infty]$ be proper, lower semicontinuous and uniformly essentially strongly convex with modulus $\gamma \in \Gamma_U$. Then*

  (i) *$f^*$ is proper and essentially Fréchet differentiable with Fréchet derivative*

$$Df^*(\xi) = \arg\max_{x \in X} \langle x, \xi \rangle - f(x). \tag{5}$$

  *If, in addition, $\tilde{\gamma}(r) := \gamma(r)/r$ is strictly increasing, then*

$$\|Df^*(\xi_1) - Df^*(\xi_2)\| \leq \tilde{\gamma}^{-1}(\|\xi_1 - \xi_2\|_*/2). \tag{6}$$

  *In other words, $Df^*$ is uniformly continuous with modulus of continuity $\chi(r) = \tilde{\gamma}^{-1}(r/2)$.*

  (ii) *$f^*$ is uniformly essentially smooth with modulus $\gamma^*$.*

**Corollary 1.** *If $\gamma(r) \geq C r^{1+\kappa}, \ \forall r \geq 0$ then $\|Df^*(\xi_1) - Df^*(\xi_2)\| \leq (2C)^{-1/\kappa} \|\xi_1 - \xi_2\|_*^{1/\kappa}$. In particular, with $\gamma(r) = \frac{K}{2} r^2$, Definition 1 becomes the classic definition of $K$-strong convexity, and (6) yields the result familiar from the finite-dimensional case that the gradient $Df^*$ is $1/K$ Lipschitz with respect to the dual norm (Nesterov, 2009, Lemma 1).*

## 2.2 Dual Averaging in Reflexive Banach Spaces

We call a proper convex function $h : X \to (-\infty, +\infty]$ a *regularizer function* on a set $\mathcal{X} \subset X$ if $h$ is essentially strongly convex and $\mathrm{dom}\, h = \mathcal{X}$. We emphasize that we do not assume $h$ to be Fréchet-differentiable. Definition 1 in conjunction with Lemma S.1 (supplemental material) implies that for any regularizer $h$, the supremum of any function of the form $\langle \cdot, \xi \rangle - h(\cdot)$ over $X$, where $\xi \in X^*$, will be attained at a unique element of $\mathcal{X}$, namely $Dh^*(\xi)$, the Fréchet gradient of $h^*$ at $\xi$.

DA with regularizer $h$ and a sequence of *learning rates* $(\eta_t)_{t \geq 1}$ generates a sequence of decisions using the simple update rule $x_{t+1} = Dh^*(\eta_t U_t)$, where $U_t = \sum_{\tau=1}^t u_\tau$ and $U_0 := 0$.

**Theorem 2.** *Let $h$ be a uniformly essentially strongly convex regularizer on $\mathcal{X}$ with modulus $\gamma$ and let $(\eta_t)_{t \geq 1}$ be a positive non-increasing sequence of learning rates. Then, for any sequence of payoff functions $(u_t)_{t \geq 1}$ in $X^*$ for which there exists $M < \infty$ such that $\sup_{x \in \mathcal{X}} |\langle u_t, x \rangle| \leq M$ for all t, the sequence of plays $(x_t)_{t \geq 0}$ given by*

$$x_{t+1} = Dh^*\big(\eta_t \textstyle\sum_{\tau=1}^t u_\tau\big) \tag{7}$$

*ensures that*

$$R_t(x) := \sum_{\tau=1}^t \langle u_\tau, x \rangle - \sum_{\tau=1}^t \langle u_\tau, x_\tau \rangle \leq \frac{h(x) - \underline{h}}{\eta_t} + \sum_{\tau=1}^t \|u_\tau\|_* \, \tilde{\gamma}^{-1}\Big(\frac{\eta_{\tau-1}}{2} \|u_\tau\|_*\Big) \tag{8}$$

*where $\underline{h} = \inf_{x \in \mathcal{X}} h(x)$, $\tilde{\gamma}(r) := \gamma(r)/r$ and $\eta_0 := \eta_1$.*

It is possible to obtain a regret bound similar to (8) also in a continuous-time setting. In fact, following Kwon and Mertikopoulos (2014), we derive the bound (8) by first proving a bound on a suitably defined notion of continuous-time regret, and then bounding the difference between the continuous-time and discrete-time regrets. This analysis is detailed in the supplementary material. Note that the condition that $\sup_{x \in \mathcal{X}} |\langle u_t, x \rangle| \leq M$ in Theorem 2 is weaker than the one in Sridharan and Tewari (2010), as it does not imply a uniformly bounded strategy set (e.g., if $X = L^2(\mathbb{R})$ and $\mathcal{X}$ is the set of distributions on $X$, then $\mathcal{X}$ is unbounded in $L^2$, but the condition may still hold).

Theorem 2 provides a regret bound for a particular choice $x \in \mathcal{X}$. Recall that $\mathcal{R}_t := \sup_{x \in \mathcal{X}} R_t(x)$. In Example 1 the set $\mathcal{X}$ is compact, so any continuous regularizer $h$ will be bounded, and hence taking the supremum over $x$ in (8) poses no issue. However, this is not the case in our general setting, as the regularizer may be unbounded on $\mathcal{X}$. For instance, consider Example 2 with the entropy regularizer $h(x) = \int_S x(s) \log(x(s)) ds$, which is easily seen to be unbounded on $\mathcal{X}$. As a consequence, obtaining a worst-case bound will in general require additional assumptions on the reward functions and the decision set $\mathcal{X}$. This will be investigated in detail in Section 3.

**Corollary 2.** *Suppose that $\gamma(r) \geq C\, r^{1+\kappa}, \ \forall\, r \geq 0$ for some $C > 0$ and $\kappa > 0$. Then*

$$R_t(x) \leq \frac{h(x) - \underline{h}}{\eta_t} + (2C)^{-1/\kappa} \sum_{\tau=1}^t \eta_{\tau-1}^{1/\kappa} \|u_\tau\|_*^{1+1/\kappa}. \tag{9}$$

*In particular, if $\|u_t\|_* \leq M$ for all t and $\eta_t = \eta\, t^{-\beta}$, then*

$$R_t(x) \leq \frac{h(x) - \underline{h}}{\eta} t^\beta + \frac{\kappa}{\kappa - \beta}\Big(\frac{\eta}{2C}\Big)^{1/\kappa} M^{1+1/\kappa}\, t^{1-\beta/\kappa}. \tag{10}$$

Assuming $h$ is bounded, optimizing over $\beta$ yields a rate of $R_t(x) = \mathcal{O}(t^{\frac{\kappa}{1+\kappa}})$. In particular, if $\gamma(r) = \frac{K}{2} r^2$, which corresponds to the classic definition of strong convexity, then $R_t(x) = \mathcal{O}(\sqrt{t})$. For non-vanishing $u_\tau$ we will need that $\eta_t \searrow 0$ for the sum in (9) to converge. Thus we could get potentially tighter control over the rate of this term for $\kappa < 1$, at the expense of larger constants.

## 3 Online Optimization on Compact Metric Spaces

We now apply the above results to the problem minimizing regret on compact metric spaces under the additional assumption of uniformly continuous reward functions. We make no assumptions on convexity of either the feasible set or the rewards. Essentially, we lift the non-convex problem of minimizing a sequence of functions over the (possibly non-convex) set $S$ to the convex (albeit infinite-dimensional) problem of minimizing a sequence of linear functionals over a set $\mathcal{X}$ of probability measures (a convex subset of the vector space of measures on $S$).

### 3.1 An Upper Bound on the Worst-Case Regret

Let $(S, d)$ be a compact metric space, and let $\mu$ be a Borel measure on $S$. Suppose that the reward vectors $u_\tau$ are given by elements in $L^q(S, \mu)$, where $q > 1$. Let $X = L^p(S, \mu)$, where $p$ and $q$ are Hölder conjugates, i.e., $\frac{1}{p} + \frac{1}{q} = 1$. Consider $\mathcal{X} = \{x \in X : x \geq 0 \text{ a.e.}, \|x\|_1 = 1\}$, the set of probability measures on $S$ that are absolutely continuous w.r.t. $\mu$ with $p$-integrable Radon-Nikodym derivatives. Moreover, denote by $\mathcal{Z}$ the class of non-decreasing $\chi : [0, \infty) \to [0, \infty]$ such that $\lim_{r \to 0} \chi(r) = \chi(0) = 0$. The following assumption will be made throughout this section:

**Assumption 1.** *The reward vectors $u_t$ have modulus of continuity $\chi$ on $S$, uniformly in $t$. That is, there exists $\chi \in \mathcal{Z}$ such that $|u_t(s) - u_t(s')| \leq \chi(d(s, s'))$ for all $t$ and for all $s, s' \in S$.*

Let $B(s, r) = \{s' \in S : d(s, s') < r\}$ and denote by $\mathcal{B}(s, \delta) \subset \mathcal{X}$ the elements of $\mathcal{X}$ with support contained in $B(s, \delta)$. Furthermore, let $D_S := \sup_{s, s' \in S} d(s, s')$. Then we have the following:

**Theorem 3.** *Let $(S, d)$ be compact, and suppose that Assumption 1 holds. Let $h$ be a uniformly essentially strongly convex regularizer on $\mathcal{X}$ with modulus $\gamma$, and let $(\eta_t)_{t \geq 1}$ be a positive non-increasing sequence of learning rates. Then, under (7), for any positive sequence $(\vartheta_t)_{t \geq 1}$,*

$$\mathcal{R}_t \leq \frac{\sup_{s \in S} \inf_{x \in \mathcal{B}(s, \vartheta_t)} h(x) - \underline{h}}{\eta_t} + t\,\chi(\vartheta_t) + \sum_{\tau=1}^{t} \|u_\tau\|_* \, \tilde{\gamma}^{-1}\left(\frac{\eta_{\tau-1}}{2} \|u_\tau\|_*\right). \qquad (11)$$

**Remark 1.** *The sequence $(\vartheta_t)_{t \geq 1}$ in Theorem 3 is not a parameter of the algorithm, but rather a parameter in the regret bound. In particular, (11) holds true for any such sequence, and we will use this fact later on to obtain explicit bounds by instantiating (11) with a particular choice of $(\vartheta_t)_{t \geq 1}$.*

It is important to realize that the infimum over $\mathcal{B}(s, \vartheta_t)$ in (11) may be infinite, in which case the bound is meaningless. This happens for example if $s$ an isolated point of some $S \subset \mathbb{R}^n$ and $\mu$ is the Lebesgue measure, in which case $\mathcal{B}(s, \vartheta_t) = \emptyset$. However, under an additional regularity assumption on the measure $\mu$ we can avoid such degenerate situations.

**Definition 4** (Heinonen. et al., 2015). *A Borel measure $\mu$ on a metric space $(S, d)$ is (Ahlfors) $Q$-regular if there exist $0 < c_0 \leq C_0 < \infty$ such that for any open ball $B(s, r)$*

$$c_0 r^Q \leq \mu(B(s, r)) \leq C_0 r^Q. \qquad (12)$$

*We say that $\mu$ is $r_0$-locally $Q$-regular if (12) holds for all $0 < r \leq r_0$.*

Intuitively, under an $r_0$-locally $Q$-regular measure, the mass in the neighborhood of any point of $S$ is uniformly bounded from above and below. This will allow, at each iteration $t$, to assign sufficient probability mass around the maximizer(s) of the cumulative reward function.

**Example 3.** *The canonical example for a $Q$-regular measure is the Lebesgue measure $\lambda$ on $\mathbb{R}^n$. If $d$ is the metric induced by the Euclidean norm, then $Q = n$ and the bound (12) is tight with $c_0 = C_0$, a dimensional constant. However, for general sets $S \subset \mathbb{R}^n$, $\lambda$ need not be locally $Q$-regular. A sufficient condition for local regularity of $\lambda$ is that $S$ is $v$-uniformly fat (Krichene et al., 2015).*

**Assumption 2.** *The measure $\mu$ is $r_0$-locally $Q$-regular on $(S, d)$.*

Under Assumption 2, $\mathcal{B}(s, \vartheta_t) \neq \emptyset$ for all $s \in S$ and $\vartheta_t > 0$, hence we may hope for a bound on $\inf_{x \in \mathcal{B}(s, \vartheta_t)} h(x)$ uniform in $s$. To obtain explicit convergence rates, we have to consider a more specific class of regularizers.

### 3.2 Explicit Rates for $f$-Divergences on $L^p(S)$

We consider a particular class of regularizers called $f$-divergences or Csiszár divergences (Csiszár, 1967). Following Audibert et al. (2014), we define $\omega$-potentials and the associated $f$-divergence.

**Definition 5.** *Let $\omega \leq 0$ and $a \in (-\infty, +\infty]$. A continuous increasing diffeomorphism $\phi : (-\infty, a) \to (\omega, \infty)$, is an $\omega$-potential if $\lim_{z \to -\infty} \phi(z) = \omega$, $\lim_{z \to a} \phi(z) = +\infty$ and $\phi(0) \leq 1$. Associated to $\phi$ is the convex function $f_\phi : [0, \infty) \to \mathbb{R}$ defined by $f_\phi(x) = \int_1^x \phi^{-1}(z)\, dz$ and the $f_\phi$-divergence, defined by $h_\phi(x) = \int_S f_\phi\big(x(s)\big)\, d\mu(s) + \iota_{\mathcal{X}}(x)$, where $\iota_{\mathcal{X}}$ is the indicator function of $\mathcal{X}$ (i.e. $\iota_{\mathcal{X}}(x) = 0$ if $x \in \mathcal{X}$ and $\iota_{\mathcal{X}}(x) = +\infty$ if $x \notin \mathcal{X}$).*

A remarkable fact is that for regularizers based on $\omega$ potentials, the DA update (7) can be computed efficiently. More precisely, it can be shown (see Proposition 3 in Krichene (2015)) that the maximizer in this case has a simple expression in terms of the dual problem, and the problem of computing $x_{t+1} = Dh^*(\eta_t \sum_{\tau=1}^{t} u_\tau)$ reduces to computing a scalar dual variable $\nu_t^*$.

**Proposition 1.** *Suppose that $\mu(S) = 1$, and that Assumption 2 holds with constants $r_0 > 0$ and $0 < c_0 \leq C_0 < \infty$. Under the Assumptions of Theorem 3, with $h = h_\phi$ the regularizer associated to an $\omega$-potential $\phi$, we have that, for any positive sequence $(\vartheta_t)_{t \geq 1}$ with $\vartheta_t \leq r_0$,*

$$\frac{\mathcal{R}_t}{t} \leq \frac{\min(C_0 \vartheta_t^Q, \mu(S))}{t \, \eta_t} f_\phi\big(c_0^{-1} \vartheta_t^{-Q}\big) + \chi(\vartheta_t) + \frac{1}{t} \sum_{\tau=1}^t \|u_\tau\|_* \, \tilde{\gamma}^{-1}\Big(\frac{\eta_{\tau-1}}{2} \|u_\tau\|_*\Big). \qquad (13)$$

For particular choices of the sequences $(\eta_t)_{t \geq 1}$ and $(\vartheta_t)_{t \geq 1}$, we can derive explicit regret rates.

### 3.3 Analysis for Entropy Dual Averaging (The Generalized Hedge Algorithm)

Taking $\phi(z) = e^{z-1}$, we have that $f_\phi(x) = \int_1^x \phi^{-1}(z) dz = x \log x$, and hence the regularizer is $h_\phi(x) = \int_S x(s) \log x(s) d\mu(s)$. Then $Dh^*(\xi)(s) = \frac{\exp \xi(s)}{\|\exp \xi(s)\|_1}$. This corresponds to a generalized Hedge algorithm (Arora et al., 2012; Krichene et al., 2015) or the entropic barrier of Bubeck and Eldan (2014) for Euclidean spaces. The regularizer $h_\phi$ can be shown to be essentially strongly convex with modulus $\gamma(r) = \frac{1}{2} r^2$.

**Corollary 3.** *Suppose that $\mu(S) = 1$, that $\mu$ is $r_0$-locally $Q$-regular with constants $c_0, C_0$, that $\|u_t\|_* \leq M$ for all $t$, and that $\chi(r) = C_\alpha r^\alpha$ for $0 < \alpha \leq 1$ (that is, the rewards are $\alpha$-Hölder continuous). Then, under Entropy Dual Averaging, choosing $\eta_t = \eta \sqrt{\log t / t}$ with $\eta = \frac{1}{M}\big(\frac{C_0 Q}{2c_0} \log(c_0^{-1} \vartheta^{-Q/\alpha}) + \frac{Q}{2\alpha}\big)^{1/2}$ and $\vartheta > 0$, we have that*

$$\frac{\mathcal{R}_t}{t} \leq \left(2M \sqrt{\frac{2C_0}{c_0}\Big(\log(c_0^{-1}\vartheta^{-Q/\alpha}) + \frac{Q}{2\alpha}\Big)} + C_\alpha \vartheta\right) \sqrt{\frac{\log t}{t}} \qquad (14)$$

*whenever $\sqrt{\log t / t} < r_0^\alpha \vartheta^{-1}$.*

One can now further optimize over the choice of $\vartheta$ to obtain the best constant in the bound. Note also that the case $\alpha = 1$ corresponds to Lipschitz continuity.

### 3.4 A General Lower Bound

**Theorem 4.** *Let $(S, d)$ be compact, suppose that Assumption 2 holds, and let $w : \mathbb{R} \to \mathbb{R}$ be any function with modulus of continuity $\chi \in \mathcal{Z}$ such that $\|w(d(\,\cdot\,, s'))\|_q \leq M$ for some $s' \in S$ for which there exists $s \in S$ with $d(s, s') = D_S$. Then for any online algorithm, there exist a sequence $(u_\tau)_{\tau=1}^t$ of reward vectors $u_\tau \in X^*$ with $\|u_\tau\|_* \leq M$ and modulus of continuity $\chi_\tau < \chi$ such that*

$$\mathcal{R}_t \geq \frac{w(D_S)}{2\sqrt{2}} \sqrt{t}, \qquad (15)$$

Maximizing the constant in (15) is of interest in order to benchmark the bound against the upper bounds obtained in the previous sections. This problem is however quite challenging, and we will defer this analysis to future work. For Hölder-continuous functions, we have the following result:

**Proposition 2.** *In the setting of Theorem 4, suppose that $\mu(S) = 1$ and that $\chi(r) = C_\alpha r^\alpha$ for some $0 < \alpha \leq 1$. Then*

$$\mathcal{R}_t \geq \frac{\min\big(C_\alpha^{1/\alpha} D_S^\alpha \,, M\big)}{2\sqrt{2}} \sqrt{t}. \qquad (16)$$

Observe that, up to a $\sqrt{\log t}$ factor, the asymptotic rate of this general lower bound for any online algorithm matches that of the upper bound (14) of Entropy Dual Averaging.

## 4 Learning in Continuous Two-Player Zero-Sum Games

Consider a two-player zero sum game $\mathcal{G} = (S_1, S_2, u)$, in which the strategy spaces $S_1$ and $S_2$ of player 1 and 2, respectively, are Hausdorff spaces, and $u : S_1 \times S_2 \to \mathbb{R}$ is the payoff function of player 1 (as $\mathcal{G}$ is zero-sum, the payoff function of player 2 is $-u$). For each $i$, denote by $\mathcal{P}_i := \mathcal{P}(S_i)$ the set of Borel probability measures on $S_i$. Denote $S := S_1 \times S_2$ and $\mathcal{P} := \mathcal{P}_1 \times \mathcal{P}_2$. For a (joint) mixed strategy $x \in \mathcal{P}$, we define the natural extension $\bar{u} : \mathcal{P} \to \mathbb{R}$ by $\bar{u}(x) := \mathbb{E}_x[u] = \int_S u(s^1, s^2) \, dx(s^1, s^2)$, which is the expected payoff of player 1 under $x$.

A continuous zero-sum game $\mathcal{G}$ is said to have *value V* if

$$\sup_{x^1 \in \mathcal{P}_1} \inf_{x^2 \in \mathcal{P}_2} \bar{u}(x^1, x^2) = \inf_{x^2 \in \mathcal{P}_2} \sup_{x^1 \in \mathcal{P}_1} \bar{u}(x^1, x^2) = V. \tag{17}$$

The elements $x^1 \times x^2 \in \mathcal{P}$ at which (17) holds are the (mixed) Nash Equilibria of $\mathcal{G}$. We denote the set of Nash equilibria of $\mathcal{G}$ by $\mathcal{N}(\mathcal{G})$. In the case of finite games, it is well known that every two-player zero-sum game has a value. This is not true in general for continuous games, and additional conditions on strategy sets and payoffs are required, see e.g. (Glicksberg, 1950).

## 4.1 Repeated Play

We consider repeated play of the continuous two-player zero-sum game. Given a game $\mathcal{G}$ and a sequence of plays $(s_t^1)_{t \geq 1}$ and $(s_t^2)_{t \geq 1}$, we say that player $i$ has sublinear (realized) regret if

$$\limsup_{t \to \infty} \frac{1}{t} \left( \sup_{s^i \in S_i} \sum_{\tau=1}^t u_i(s^i, s_\tau^{-i}) - \sum_{\tau=1}^t u_i(s_\tau^i, s_\tau^{-i}) \right) \leq 0 \tag{18}$$

where we use $-i$ to denote the other player.

A strategy $\sigma^i$ for player $i$ is, loosely speaking, a (possibly random) mapping from past observations to its actions. Of primary interest to us are Hannan-consistent strategies:

**Definition 6** (Hannan, 1957). *A strategy $\sigma^i$ of player $i$ is Hannan consistent if, for any sequence $(s_{-i}^t)_{t \geq 1}$, the sequence of plays $(s_i^t)_{t \geq 1}$ generated by $\sigma^i$ has sublinear regret almost surely.*

Note that the almost sure statement in Definition 6 is with respect to the randomness in the strategy $\sigma^i$. The following result is a generalization of its counterpart for discrete games (e.g. Corollary 7.1 in (Cesa-Bianchi and Lugosi, 2006)):

**Proposition 3.** *Suppose $\mathcal{G}$ has value $V$ and consider a sequence of plays $(s_t^1)_{t \geq 1}$, $(s_t^2)_{t \geq 1}$ and assume that both players have sublinear realized regret. Then $\lim_{t \to \infty} \frac{1}{t} \sum_{\tau=1}^t u(s_\tau^1, s_\tau^2) = V$.*

As in the discrete case (Cesa-Bianchi and Lugosi, 2006), we can also say something about convergence of the empirical distributions of play to the set of Nash Equilibria. Since these distributions have finite support for every $t$, we can at best hope for convergence in the weak sense as follows:

**Theorem 5.** *Suppose that in a repeated two-player zero sum game $\mathcal{G}$ that has a value both players follow a Hannan-consistent strategy, and denote by $\hat{x}_t^i = \frac{1}{t} \sum_{\tau=1}^t \delta_{s_\tau^i}$ the marginal empirical distribution of play of player $i$ at iteration $t$. Let $\hat{x}_t := (\hat{x}_t^1, \hat{x}_t^2)$. Then $\hat{x}_t \rightharpoonup \mathcal{N}(\mathcal{G})$ almost surely, that is, with probability 1 the sequence $(\hat{x}_t)_{t \geq 1}$ weakly converges to the set of Nash equilibria of $\mathcal{G}$.*

**Corollary 4.** *If $\mathcal{G}$ has a unique Nash equilibrium $x^*$, then with probability 1, $\hat{x}_t \rightharpoonup x^*$.*

## 4.2 Hannan-Consistent Strategies

By Theorem 5, if each player follows a Hannan-consistent strategy, then the empirical distributions of play weakly converge to the set of Nash equilibria of the game. But do such strategies exist? Regret minimizing strategies are intuitive candidates, and the intimate connection between regret minimization and learning in games is well studied in many cases, e.g. for finite games (Cesa-Bianchi and Lugosi, 2006) or potential games (Monderer and Shapley, 1996). Using our results from Section 3, we will show that, under the appropriate assumption on the information revealed to the player, no-regret learning based on Dual Averaging leads to Hannan consistency in our setting.

Specifically, suppose that after each iteration $t$, each player $i$ observes a *partial payoff function* $\tilde{u}_t^i : S_i \to \mathbb{R}$ describing their payoff as a function of only their own action, $s_i$, holding the action played by the other player fixed. That is, $\tilde{u}_t^1(s^1) := u(s^1, s_t^2)$ and $\tilde{u}_t^2(s^2) := -u(s_t^1, s^2)$.

**Remark 2.** *Note that we do not assume that the players have knowledge of the joint utility function $u$. However, we do assume that the player has full information feedback, in the sense that they observe partial reward functions $u(\,\cdot\,, s_\tau^{-i})$ on their entire action set, as opposed to only observing the reward $u(s_\tau^1, s_\tau^2)$ of the action played (the latter corresponds to the bandit setting).*

We denote by $\tilde{U}_t^i = (\tilde{u}_\tau^i)_{\tau=1}^t$ the sequence of partial payoff functions observed by player $i$. We use $\mathcal{U}_t^i$ to denote the set of all possible such histories, and define $\mathcal{U}_0^i := \emptyset$. A strategy $\sigma^i$ of player $i$ is a collection $(\sigma_t^i)_{t=1}^\infty$ of (possibly random) mappings $\sigma_t^i : \mathcal{U}_{t-1}^i \to S_i$, such that at iteration $t$, player $i$ plays $s_t^i = \sigma_t^i(U_{t-1}^i)$. We make the following assumption on the payoff function:

**Assumption 3.** *The payoff function $u$ is uniformly continuous in $s^i$ with modulus of continuity independent of $s^{-i}$ for $i = 1, 2$. That is, for each $i$ there exists $\chi^i \in \mathcal{Z}$ such that $|u(s, s^{-i}) - u(s', s^{-i})| \leq \chi^i(d_i(s, s'))$ for all $s^{-i} \in S_{-i}$.*

It is easy to see that Assumption 3 implies that the game has a value (see supplementary material). It also makes our setting compatible with that of Section 3. Suppose now that each player randomizes their play according to the sequence of probability distributions on $S_i$ generated by DA with regularizer $h_i$. That is, suppose that each $\sigma_t^i$ is a random variable with the following distribution:

$$\sigma_t^i \sim Dh_i^*\left(\eta_{t-1} \sum_{\tau=1}^{t-1} \tilde{u}_\tau^i\right). \tag{19}$$

**Theorem 6.** *Suppose that player $i$ uses strategy $\sigma^i$ according to (19), and that the DA algorithm ensures sublinear regret (i.e. $\limsup_t \mathcal{R}_t / t \leq 0$). Then $\sigma^i$ is Hannan-consistent.*

**Corollary 5.** *If both players use strategies according to (19) with the respective Dual Averaging ensuring that $\limsup_t \mathcal{R}_t / t \leq 0$, then with probability 1 the sequence $(\hat{x}_t)_{t \geq 1}$ of empirical distributions of play weakly converges to the set of Nash equilibria of $\mathcal{G}$.*

**Example** Consider a zero-sum game $\mathcal{G}_1$ between two players on the unit interval with payoff function $u(s^1, s^2) = s^1 s^2 - a^1 s^1 - a^2 s^2$, where $a^1 = \frac{e-2}{e-1}$ and $a^2 = \frac{1}{e-1}$. It is easy to verify that the pair $(x^1, x^2) = \left(\frac{\exp(s)}{e-1}, \frac{\exp(1-s)}{e-1}\right)$ is a mixed-strategy Nash equilibrium of $\mathcal{G}_1$. For sequences $(s_\tau^1)_{\tau=1}^t$ and $(s_\tau^2)_{\tau=1}^t$, the cumulative payoff functions for fixed action $s \in [0, 1]$ are given, respectively, by

$$U_t^1(s^1) = \left(\Sigma_{\tau=1}^t s_\tau^2 - a^1 t\right) s^1 - a^2 \Sigma_{\tau=1}^t s_\tau^2 \qquad U_t^2(s^2) = \left(a^2 t - \Sigma_{\tau=1}^t s_\tau^1\right) s^2 - a^1 \Sigma_{\tau=1}^t s_\tau^1$$

If each player $i$ uses the Generalized Hedge Algorithm with learning rates $(\eta_\tau)_{\tau=1}^t$, their strategy in period $t$ is to sample from the distribution $x_t^i(s) \propto \exp(\alpha_t^i s)$, where $\alpha_t^1 = \eta_t(\Sigma_{\tau=1}^t s_\tau^2 - a^1 t)$ and $\alpha_t^2 = \eta_t(a^2 t - \Sigma_{\tau=1}^t s_\tau^1)$. Interestingly, in this case the sum of the opponent's past plays is a sufficient statistic, in the sense that it completely determines the mixed strategy at time $t$.

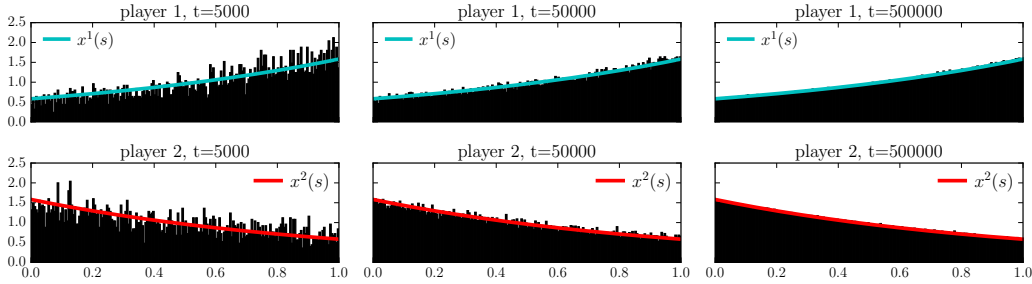

Figure 1: Normalized histograms of the empirical distributions of play in $\mathcal{G}$ (100 bins)

Figure 1 shows normalized histograms of the empirical distributions of play at different iterations $t$. As $t$ grows the histograms approach the equilibrium densities $x^1$ and $x^2$, respectively. However, this does not mean that the individual strategies $x_t^i$ converge. Indeed, Figure 2 shows the $\alpha_t^i$ oscillating around the equilibrium parameters 1 and $-1$, respectively, even for very large $t$. We do, however, observe that the time-averaged parameters $\bar{\alpha}_t^i$ converge to the equilibrium values 1 and $-1$.

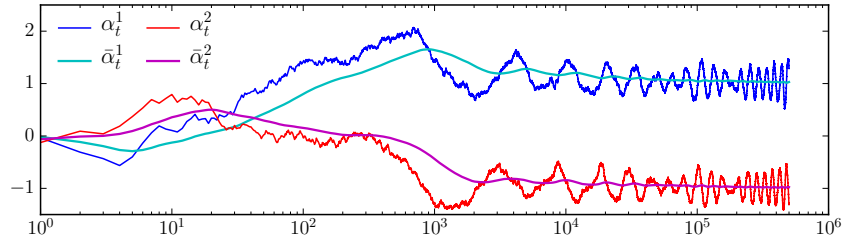

Figure 2: Evolution of parameters $\alpha_t^i$ and $\bar{\alpha}_t^i := \frac{1}{t} \sum_{\tau=1}^t \alpha_\tau^i$ in $\mathcal{G}_1$

In the supplementary material we provide additional numerical examples, including one that illustrates how our algorithms can be utilized as a tool to compute approximate Nash equilibria in continuous zero-sum games on non-convex domains.

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
