[Supplementary Material]

# Minimizing Regret on Reflexive Banach Spaces and Nash Equilibria in Continuous Zero-Sum Games – Supplementary Material

**Maximilian Balandat,   Walid Krichene,   Claire Tomlin,   Alexandre Bayen**
Electrical Engineering and Computer Sciences, UC Berkeley
[balandat,walid,tomlin]@eecs.berkeley.edu, bayen@berkeley.edu

## Contents

## S.1    Some Results From Convex Analysis

In this section we collect some results from infinite-dimensional convex analysis that will play an important role in our analysis of the Dual Averaging algorithm.

**Lemma S.1** (Asplund, 1968). *Let $f : X \to (-\infty, +\infty]$ be proper lower semicontinuous. For a pair $(x_0, \xi_0) \in X \times X^*$ the following are equivalent:*

*(i)  $f^*$ is finite and Fréchet differentiable at $\xi_0$ with Fréchet derivative $Df^*(\xi_0) = x_0$.*

*(ii)  For some $\gamma^* \in \Gamma_L$,*
$$f^*(\xi) \le f^*(\xi_0) + \langle x_0, \xi - \xi_0 \rangle + \gamma^*(\|\xi - \xi_0\|), \quad \forall \xi \in X^* \tag{1}$$
*and $f^*(\xi_0) \in \mathbb{R}$.*

*(iii)  For some $\gamma \in \Gamma_U$,*
$$f(x) \ge f(x_0) + \langle x - x_0, \xi_0 \rangle + \gamma(\|x - x_0\|), \quad \forall x \in X \tag{2}$$
*and $f(x_0) \in \mathbb{R}$.*

*(iv)  $f^*$ is finite at $\xi_0$, $\mathrm{dom}\, f^*$ is radial at $\xi_0$, and $x_j \to x_0$ in norm whenever*
$$\lim_{j \to \infty} \langle x_j, \xi_0 \rangle - f(x_j) = f^*(\xi_0) \tag{3}$$

*Any of the above conditions implies that $\langle x_0, \xi_0 \rangle = f(x_0) + f^*(\xi_0)$ (in other words: the Fenchel-Young inequality holds with equality) and that $f(x_0) = f^{**}(x_0)$. The functions $\gamma$ and $\gamma^*$ in (ii) and (iii) form a pair of mutually dual functions.*

Note that the function $f$ in Lemma S.1 need not be convex. The following result will be essential to our analysis:

**Theorem S.1** (Strömberg, 2011). *Let $f : X \to (-\infty, +\infty]$ be lower semicontinuous. Then $f^*$ is proper and essentially Fréchet differentiable if and only if $f$ is a convex proper function that is essentially strongly convex.*

## S.2    Dual Averaging in Continuous Time

In this section we use ideas from Kwon and Mertikopoulos (2014) and introduce a continuous-time regret minimization problem related to the one in discrete-time discussed in Section 2.2. In fact, this analysis will be crucial in proving the discrete-time regret bound (7) in Theorem 2.

### S.2.1    Regret Minimization in Continuous Time on Reflexive Banach Spaces

Consider a reflexive Banach space $X$ with dual $X^*$ and regularizer $h$ on $\mathcal{X}$. Furthermore, suppose that $u^c : [0, \infty) \to X^*$ is a continuous-time reward process satisfying the following assumptions:

**Assumption S.1.** *The reward process $u^c$ is locally integrable for any $x \in X$. That is, for all $x \in X$, $r_x : t \mapsto \langle u_t^c, x \rangle$ is Lebesgue-integrable on any compact set $K \subset [0, \infty)$.*

**Assumption S.2.** *There exists $M < \infty$ such that $\sup_{x \in \mathcal{X}} |\langle u_t^c, x \rangle| \le M$ for all $t$.*

Let $\eta^c : [0, \infty) \to (0, \infty)$ be a non-increasing and piece-wise continuous learning rate process. Furthermore, let $U_t^c = \int_0^t u_\tau^c \, d\tau$ be the cumulative reward function. We consider the continuous-time process $x^c : [0, \infty) \to X$ given by
$$x_t^c := Dh^*(\eta_t^c U_t^c) \tag{4}$$

**Theorem S.2** (Continuous-Time Regret Bound). *Let $h$ be a regularizer function on $\mathcal{X}$, let $\eta^c$ be non-decreasing and locally piecewise continuous. Suppose that the reward process $u^c$ satisfies Assumptions S.1 and S.2. Then under (4) we have, for any $x \in X$, that*
$$R_t^c(x) := \int_0^t \langle u_\tau^c, x \rangle \, d\tau - \int_0^t \langle u_\tau^c, x_\tau^c \rangle \, d\tau \le \frac{h(x) - \underline{h}}{\eta_t^c} \tag{5}$$
*where $\underline{h} := \inf_{x \in \mathcal{X}} h(x)$.*

*Theorem S.2.* Let $y_t^c = \eta_t^c \int_0^t u_\tau^c \, d\tau$. By linearity,

$$\eta_t^c \int_0^t \langle u_\tau^c, x \rangle \, d\tau = \eta_t^c \big\langle \int_0^t u_\tau^c \, d\tau, x \big\rangle = \langle y_t^c, x \rangle$$

Assume for now that $\eta^c \in C^1$. If $h$ is proper, then

$$\int_0^t \langle u_\tau^c, x \rangle \, d\tau = \frac{\langle y_t^c, x \rangle}{\eta_t^c} \leq \frac{h^*(y_t^c) + h(x)}{\eta_t^c} = \frac{h^*(y_t^c)}{\eta_t^c} + \frac{h(x)}{\eta_t^c} \tag{6}$$

by the Fenchel-Young inequality. By Theorem 1, $h^*$ is essentially Fréchet differentiable with Fréchet gradient $Dh^*(y)$. Furthermore, $y_t^c$ is differentiable. Thus, applying the chain rule and using that $x_t^c = Dh^*(y_t^c) = \arg\max_{x \in \mathcal{X}} \big( \langle x, y_t^c \rangle - h(x) \big) = \arg\max_{x \in X} \big( \langle x, y_t^c \rangle - h(x) \big)$ we obtain

$$\frac{d}{dt} \frac{h^*(y_t^c)}{\eta_t^c} = \frac{\eta_t^c \langle Dh^*(y_t^c), \frac{d}{dt} y_t^c \rangle - h^*(y_t^c) \, \dot{\eta}^c(t)}{(\eta_t^c)^2}$$

$$= \frac{\big\langle x_t^c, \big( \eta_t^c u_t^c + \dot{\eta}_t^c \int_0^t u^c(s, \tau) \, d\tau \big) \big\rangle}{\eta_t^c} - \frac{\dot{\eta}_t^c}{(\eta_t^c)^2} h^*(y_t^c)$$

$$= \langle x_t^c, u_t^c \rangle + \frac{\dot{\eta}_t^c}{(\eta_t^c)^2} \big( \langle x_t^c, y_t^c \rangle - h^*(y_t^c) \big)$$

$$= \langle x_t^c, u_t^c \rangle + \frac{\dot{\eta}_t^c}{(\eta_t^c)^2} h(x_t^c)$$

Now $\dot{\eta}_t^c \leq 0$ by assumption, and hence

$$\frac{d}{dt} \frac{h^*(y_t^c)}{\eta_t^c} \leq \langle x_t^c, u_t^c \rangle + \frac{\dot{\eta}_t^c}{(\eta_t^c)^2} \underline{h}$$

Integrating from $t = 0$ to $t = t$ yields

$$\frac{h^*(y_t^c)}{\eta_t^c} - \frac{h^*(y_0^c)}{\eta^c(0)} \leq \int_0^t \langle x_\tau^c, u_\tau^c \rangle \, d\tau + \underline{h} \int_0^t \frac{\dot{\eta}_\tau^c}{(\eta_\tau^c)^2} \, d\tau$$

$$= \int_0^t \langle x_\tau^c, u_\tau^c \rangle \, d\tau - \underline{h} \left( \frac{1}{\eta_t^c} - \frac{1}{\eta_0^c} \right)$$

Now $y_0^c = 0$, and hence $h^*(y_0^c) = \sup_{x \in X} -h(x) = -\underline{h}$, and so

$$\frac{h^*(y_t^c)}{\eta_t^c} \leq \int_0^t \langle x_\tau^c, u_\tau^c \rangle \, d\tau - \frac{\underline{h}}{\eta_t^c}$$

Plugging this into (6), collecting terms and rearranging yields (5).

Now suppose that $\eta^c$ is only piecewise continuous. Then there exists a sequence $(\eta^{c,i})_{i=1}^\infty$ of positive nonincreasing $C^1$ functions such that $\eta^{c,i} \to \eta^c$ pointwise a.e.. Let $x_t^{c,i} := Dh^*(\eta_t^{c,i} U_t^c)$. Note that $Dh^*$ is continuous by Theorem 1 and thus $x_t^{c,i} \to x_t^c$ pointwise. By Assumption S.2 we have that $|\langle u_\tau^c, x^{c,i} \rangle| < M$ for all $\tau, i$ and thus $\int_0^t \langle u_\tau^c, x_\tau^{c,i} \rangle \, d\tau \to \int_0^t \langle u_\tau^c, x_\tau^c \rangle \, d\tau$ by Dominated Convergence. $\qquad \square$

### S.2.2 Online Optimization in Continuous Time on Compact Metric Spaces

One can also obtain bounds on the regret in continuous time by using similar arguments as in Section 3. While we do not make use of them in the main part of this article, these bounds may be of independent interest.

We consider the setting of Section 3. Specifically, let $(S, d)$ be a compact metric space, and let $\mu \in \mathcal{P}$, the set of Borel measures on $S$. Denote by $B(s, r) = \{s' \in S : d(s, s') < r\}$ the open ball of radius $r$ centered at $s$. For $p > 1$ consider $X = L^p(S, \mu)$ and $\mathcal{X} = \{x \in X : x \geq 0 \text{ a.e.}, \|x\|_1 = 1\}$, the set of probability measures on $S$ that are absolutely continuous w.r.t. $\mu$ and whose Radon-Nikodym densities are $p$-integrable. Denote by $D_S := \sup_{s,s' \in S} d(s, s')$ the diameter of $S$ and by $\mathcal{B}(s, \vartheta_t^c) \subset \mathcal{X}$ the set of elements of $\mathcal{X}$ with support contained in $B(s, \vartheta_t^c)$. We need the following continuous-time variant of Assumption 1:

**Assumption S.3.** *The reward process $u^c$ has modulus of continuity $\chi$ on $S$, uniformly in $t$. That is, there exists $\chi \in \mathcal{Z}$ such that $|u_t^c(s) - u_t^c(s')| \leq \chi(d(s, s'))$ for all $s, s' \in S$ for all $t$.*

**Theorem S.3** (Continuous-Time Regret Bound on Metric Spaces). *Let $(S, d)$ be compact, and suppose that Assumption S.3 holds. Let $h$ be a regularizer function on $\mathcal{X}$, and let $\eta^c$ be non-decreasing and locally piecewise continuous. Suppose further that $\vartheta^c : [0, \infty) \to (0, \infty)$ is a non-negative function and that the reward process $u^c$ satisfies Assumptions S.1 and S.2. Then, under the process* (4),

$$\mathcal{R}_t^c \leq \frac{\sup_{s \in S} \inf_{x \in \mathcal{B}(s, \vartheta_t^c)} h(x)}{\eta_t^c} + t\,\chi(\vartheta_t^c) - \frac{h}{\eta_t^c} \tag{7}$$

*Theorem S.3.* Similar to the proof of Theorem 3. □

**Proposition S.1.** *Suppose that Assumption 2 holds with constants $c_0 > 0$ and $C_0 < \infty$. Under the Assumptions of Theorem S.3, with essentially strongly convex regularizer $h_\phi$ the $f$-divergence of an $\omega$-potential $\phi$, we have the following regret bound:*

$$\frac{\mathcal{R}_t^c}{t} \leq \frac{\min(C_0\,(\vartheta_t^c)^Q, \mu(S))}{t\,\eta_t^c} f_\phi\big(c_0^{-1}(\vartheta_t^c)^{-Q}\big) + \chi(\vartheta_t^c) \tag{8}$$

*Proposition S.1.* Similar to the proof of Proposition 1. □

### S.2.3 Consistency of Dual Averaging

It is quite intuitive to see that Dual Averaging would recover the greedy algorithm as the regularizer $h$ "approaches a constant". In this section, we make this intuition precise.

**Definition S.1** (Consistency of a Sequence of Regularizers). *A sequence $(h_1, h_2, \dots)$ of regularizers on $\mathcal{X}$ is consistent if there exists $C \in \mathbb{R}$ such that $h_i(x) \to C$ as $i \to \infty$ for all $x \in \mathcal{X}$.*

For $s \in S$, $A \subset S$, let $d(s, A) = \inf_{s' \in A} d(s, s')$. For $\delta > 0$, let $B_\delta^* := \{s \in S : d(s, S^*) < \delta\}$. Moreover, let $\nu|_A$ denote the restriction of $\nu \in \mathcal{P}(S)$ to $A$.

**Proposition S.2.** *Suppose Assumption 2 holds and that $(h_i)_{i \geq 1}$ is a sequence of regularizers that is consistent. Fix $t$ and let $U^* := \max_{s \in S} U_t(s)$ and $S^* := \{s \in S : U_t(s) = U^*\}$. For $i \geq 1$ let $x_{t,i}^* := Dh_i^*(U_t)$ Then, for any $\delta > 0$, we have that $x_{t,i}^*|_{(B_\delta^*)^c} \to 0|_{(B_\delta^*)^c}$ (strongly) as $i \to \infty$. Equivalently, $\int_{S^*} x_i^*(s)\,ds \to 1$ as $i \to \infty$.*

Proposition S.2 shows that if the sequence of regularizers is consistent, the optimizers, in the limit, collapse to distributions supported on the set of maximizers of $U_t$ (as illustrated numerically in Example S.2 in section S.5 of the supplementary material). If the maximizer of $U_t$ is unique, we can say the following:

**Corollary S.1.** *In the setting of Proposition S.2, suppose that $U_t$ admits a unique maximizer $s_t^* \in S$. Then $x_i^*$ weakly converges to the Dirac measure on $s_t^*$ as $i \to \infty$. We write $x_{t,i}^* \rightharpoonup \delta_{s_t^*}$.*

## S.3 Computing the Dual Averaging Optimizer

In this section we discuss some aspects concerning the computation of the optimizer in the Dual Averaging update in the setting of online optimization on compact metric spaces with uniformly continuous rewards. The results of this section are used for generating the Hannan-consistent strategies in the repeated game in the Example in the paper as well as the examples in Section S.4, and for performing the numerical benchmarks of the algorithms in Appendix S.5.

As pointed out in Section 3.2, it can be shown that for $f$-Divergences of $\omega$-potentials, the Fréchet differential $Dh^*$ in this case has a simple expression in terms of the dual problem, and the problem of computing $x_{t+1} = Dh^*(\eta_t \sum_{\tau=1}^t u_\tau)$ reduces to computing a scalar dual variable $\nu_t^*$. In particular, one can show the following:

**Proposition S.3** (Krichene, 2015). *Let $\phi$ be an $\omega$-potential with associated $f$-Divergence $h_\phi$ on $\mathcal{X}$. Then*

$$Dh_\phi^*(\xi) = \phi(\xi + \nu^\star)_+ \tag{9}$$

*where $(\,\cdot\,)_+$ denotes the positive part of $(\,\cdot\,)$, and $\nu^\star$ satisfies $\int_S \phi(\xi + \nu^\star)_+\, d\mu(s) = 1$.*

By Proposition S.3, the Fréchet derivative $Dh_\phi^*$ at $\xi = \eta_t U_t$ is entirely determined by the dual variable $\nu^\star$, the unique $\nu$ such that $f(\nu) = 1$, where $f(\nu) = \int_S \phi(\eta_t(U_t(s) + \nu^\star))_+ \, d\mu(s)$. Since $f$ is increasing by assumption on $\phi$, $\nu^\star$ can be determined using a simple bisection method. To guide the search for $\nu_t^\star$ for $t > 0$ we can make use of the following result:

**Proposition S.4.** *Suppose $\phi$ is convex and let $\nu_t^\star$ the optimal dual variable determining $Dh_\phi^*(\eta_t U_t)$. Then*

$$\frac{\eta_t}{\eta_{t+1}}\nu_t^\star - M \;\leq\; \nu_{t+1}^\star \;\leq\; \frac{\eta_t}{\eta_{t+1}}\nu_t^\star + \frac{\eta_t - \eta_{t+1}}{\eta_{t+1}} \, tM \tag{10}$$

*where $\nu_0^\star = \eta_0^{-1}\phi^{-1}(1)$. Moreover, for $\eta_t = \eta \, t^{-\beta}$ this interval has length $\approx (1 + \beta)M$.*

*Proposition S.4.* Since $U_t \equiv 0$, we have $\nu_0^\star = \eta_0^{-1}\phi^{-1}(1)$. Moreover, by definition we have

$$\int_S \phi\big(\eta_t\big(U_t(s) + \nu_t^\star\big)\big)_+ \, d\mu(s) = \int_S \phi\big(\eta_{t+1}\big(U_{t+1}(s) + \nu_{t+1}^\star\big)\big)_+ \, d\mu(s) = 1$$

If $\phi$ is convex, then so is $\phi(\,\cdot\,)_+$ as $z \mapsto z_+$ is convex and nondecreasing. Therefore

$$1 = \int_S \phi\Big(\eta_t\big(U_t(s) + \nu_t^\star\big) + (\eta_{t+1} - \eta_t)U_t(s) - \eta_t\nu_t^\star + \eta_{t+1}\nu_{t+1}^\star + \eta_{t+1}u_{t+1}(s)\Big)_+ \, d\mu(s)$$

$$\leq \int_S \phi\Big(\eta_t\big(U_t(s) + \nu_t^\star\big)\Big)_+ + \phi\Big(\eta_t\big(U_t(s) + \nu_t^\star\big)\Big)_+' \Big((\eta_{t+1} - \eta_t)U_t(s)$$

$$- \eta_t\nu_t^\star + \eta_{t+1}\nu_{t+1}^\star + \eta_{t+1}u_{t+1}(s)\Big)d\mu(s)$$

$$\leq 1 + \int_S \phi\big(\eta_t\big(U_t(s) + \nu_t^\star\big)\big)_+' \big(\eta_{t+1}\nu_{t+1}^\star - \eta_t\nu_t^\star + \eta_{t+1}M\big)d\mu(s)$$

and hence, since $\phi' \geq 0$, we must have that $\eta_{t+1}\nu_{t+1}^\star - \eta_t\nu_t^\star + \eta_{t+1}M \geq 0$. Rearranging yields the lower bound on $\nu_{t+1}^\star$. The other inequality is proven in a similar fashion by reversing the roles of $t$ and $t + 1$. Finally, to show that the interval has length $\approx (1 + \beta)M$ independent of $t$, note that $\frac{\eta_t}{\eta_{t+1}} = (1 + \frac{1}{t})^\beta \approx 1 + \frac{\beta}{t}$, and so $\frac{\eta_t - \eta_{t+1}}{\eta_{t+1}}tM \approx \beta M$. $\qquad\square$

Having determined $\nu_t^\star$, we then have an explicit form of the distribution over $S$ from which to sample $s_{t+1}$. For this, a variety of established methods can be used, from simple rejection sampling in low dimensions (employed in our simulations) to MCMC methods (e.g. slice sampling) in higher dimensions. In cetain special cases, sampling from $x_t$ may be done very efficiently. For example, if the losses are affine, the domain $S$ is a hyperrectangle, and the potential is a generalized Exponential Potential, then $s_{t+1}$ can be obtained by sampling from $n$ independent truncated exponential random variables. The main computational challenge is then to compute the integral in $f$. Off-the-shelf numerical integration schemes work well if $n$ is small, but are typically not applicable in higher dimensions. Instead, one has to resort to other methods, such as Monte Carlo methods or sparse grids.

## S.4    Additional Examples of Learning Nash Equilibria in Continuous Games

### S.4.1    A Game With Unique Mixed Strategy Equilibrium

Consider the zero-sum game $\mathcal{G}_2$ between two players playing on the unit interval $S_i = [0, 1]$ with payoff function given by

$$u(s^1, s^2) = \frac{(1 + s^1)(1 + s^2)(1 - s^1s^2)}{(1 + s^1s^2)^2} \tag{11}$$

Since $|D_{s^i}u| \leq 8$ for any $s^{-i} \in [0, 1]$ the payoff function is Lipschitz. It can be shown that $V = 4/\pi$ and that this game has no pure and a unique mixed Nash equilibrium, with equilibrium density $x^i(s) = \frac{2}{\pi\sqrt{s}(1+s)}$ the same for both players (Glicksberg and Gross, 1953). Note that $x^i$ is unbounded and that $x^i \in L^p(S_i, \lambda)$ for any $1 \leq p < 2$. This unboundedness is the reason for the slow convergence of the empirical distributions to $x^i$ near zero that we can observe in Figure 1.

Figure 1: Normalized histograms of the empirical distributions of play for $\mathcal{G}_2$ (200 bins)

### S.4.2 A Game on a Non-Convex Domain

One of the most interesting features of the Dual Averaging algorithms discussed in Section 3 is that they are applicable also in case of non-convex domains. We may therefore utilize them as a tool to compute approximate Nash equilibria in continuous zero-sum games on non-convex domains. In particular, consider a game $\mathcal{G}_3$ in which each $S_i = [0, 2]^2 \setminus [0.4, 1]^2$ is an $L$-shaped subset of $\mathbb{R}^2$. It is easy to see that the Lebesgue measure on this set is $Q$-regular with $Q = 2$, $c_0 = \frac{\pi}{4}$ and $C_0 = \pi$.

We define the metric $\tilde{d}$ on $S_1$ between any two points $a, b \in S_i$ as the length (in the Euclidean distance) of the shortest path between $a$ and $b$ that is entirely contained in $S_i$. The payoff function $u$ is given as $u(s^1, s^2) = \tilde{d}(s^1, s^2) - \frac{1}{10}\tilde{d}(s^1, 0)$, which can be interpreted as a "hide and seek" game in which player 1 would like to get as far away from player 2 as possible, while at the same time having a preference for being near the origin. Player 2 instead wants to be as close to player 1 as possible. Intuitively, this game will not admit a pure Nash equilibrium. Given the geometry of the problem, computing a mixed Nash equilibrium (whose existence follows from a Theorem by Glicksberg (1950)) poses a challenge.

Instead, having both players play Entropy Dual Averaging on $L^p(S_i, \lambda)$, we observe in Figure 2 that they indeed incur sublinear regret, and that the empirical distributions of play do converge. Figure 3 shows Kernel Density Estimates (KDE) of $\hat{x}_t^1$ and $\hat{x}_t^2$ after $t = 7500$ iterations.

Figure 2: Time-Average Regrets (log-log scale) for Generalized Hedge in $\mathcal{G}_3$

## S.5 Numerical Results and Comparison With Other Methods

In this section, we review some algorithms for online *convex* optimization over subsets of $\mathbb{R}^n$ that have been proposed in the literature, and compare them with our Dual Averaging method for online optimization on compact metric spaces with uniformly continuous rewards from Section 3. Such algorithms are often formulated in terms of *loss functions* $\ell_\tau$, but clearly these algorithms apply just as well by setting $\ell_\tau = -u_\tau$, as long as the set $S$ is convex and the rewards are concave and satisfy the additional assumptions made by the algorithms. Table 1 summarizes the regret bounds of each method, with the corresponding assumptions on the feasible set and the loss functions.

The bound on Dual Averaging in Table 1 is obtained by assuming the regularizer to be the $f$-divergence associated to an $\omega$-potential and making an assumption on the asymptotic growth rate of the function $f_\phi$ as follows:

Figure 3: Kernel Density Estimates of $\hat{x}_t^1$ (left) and $\hat{x}_t^2$ (right) in $\mathcal{G}_3$ for $t = 7500$

**Corollary S.2.** *Suppose that $f_\phi(x) \leq C_\phi\, x^{1+\kappa}$ for some $\kappa > 0$ and $C_\phi < \infty$. Suppose further the rewards are $\alpha$-Hölder continuous, i.e. $\chi(r) = C_\alpha\, r^\alpha$, and that $h_\phi$ is uniformly essentially strongly convex with modulus $\gamma(r) = \frac{K}{2} r^2$. Then the learning rate $\eta_t = \eta\, t^{-\beta}$ with $\eta = \frac{1}{M}\left(\frac{1+\frac{\kappa}{\alpha}Q}{2+\frac{\kappa}{\alpha}Q} \frac{C_0 C_\phi}{c_0^{1+\kappa} \vartheta^{\kappa Q}}\right)^{1/2}$ and $\beta = \frac{1}{2+\frac{\kappa}{\alpha}Q}$ yields the following bound:*

$$\frac{\mathcal{R}}{t} \leq \left(2M\tilde{C}\vartheta^{-\frac{\kappa Q}{2}} + C_\alpha \vartheta^\alpha\right) t^{-\frac{1}{2+\frac{\kappa}{\alpha}Q}} \tag{12}$$

*for any $\vartheta < r_0$, where $\tilde{C} = \sqrt{\frac{2+\frac{\kappa}{\alpha}Q}{1+\frac{\kappa}{\alpha}Q} \frac{C_0 C_\phi}{c_0^{1+\kappa}}}$.*

| Algorithm | Assumptions | Parameters | Bound on $\mathcal{R}_t/t$ |
|---|---|---|---|
| GP / OGD | $\ell_t$ convex $\quad \|\nabla\ell_t\|_2 \leq G$ | $\eta_t = \frac{1}{\sqrt{t}}$ | $\left(\frac{D^2}{2} + G^2\right)t^{-1/2} - \frac{G^2}{2}t^{-1}$ |
| | $\ell_t$ $H$-strongly convex $\quad \|\nabla\ell_t\|_2 \leq G$ | $\eta_t = \frac{1}{Ht}$ | $\frac{G^2}{2H}\frac{1+\log t}{t}$ |
| FTAL / ONS | $\ell_t$ $\alpha$-exp-concave $\quad \|\nabla\ell_t\|_2 \leq G$ | $\beta = \min(\frac{1}{8GD}, \frac{\alpha}{2})$ | $64n(\alpha^{-1} + GD)\frac{1+\log t}{t}$ |
| EWOO | $\ell_t$ $\alpha$-exp-concave | $\eta_t = \alpha$ | $\frac{n}{\alpha}\frac{1+\log(1+t)}{t}$ |
| DA with $h_\phi$ | $S$ $Q$-regular with $c_0, C_0$ $u_t$ $\alpha$-Hölder-continuous $\quad \|u_t\|_* \leq M$ $\quad f_\phi(x) \leq Cx^{1+\epsilon}\ \forall x \geq 1$ | $\eta_t = \eta\, t^{-\frac{1}{2+\frac{\kappa}{\alpha}Q}}$ | $\left(2M\tilde{C}\vartheta^{-\frac{\kappa Q}{2}} + C_\alpha\vartheta^\alpha\right)t^{-\frac{1}{2+n\kappa}}$ |

Table 1: Regret bounds of different online optimization algorithms

### S.5.1 Optimizing Sequences of Convex Functions over Convex Sets

Zinkevich (2003) formalized the online convex optimization problem, in which the feasible set $S$ and the loss functions are assumed to be convex. He proposed a Greedy Projection method (GP), summarized in Algorithm 1, which we will also refer to as Online Gradient Descent (OGD). Theorem 1 in (Zinkevich, 2003) shows that when $\|\nabla\ell_t\|$ is uniformly bounded, the regret of GP with learning rates $\eta_t = 1/\sqrt{t}$ grows as $\mathcal{O}(\sqrt{t})$. Hazan et al. (2007) show that it is possible to obtain logarithmic regret under additional assumptions on the loss functions. In particular, if the losses are $H$-strongly convex then GP with learning rates $\eta_t = \frac{1}{Ht}$ has regret $\mathcal{R}_t \leq \frac{M^2}{2H}\left(1 + \log t\right)$. They also propose methods for uniformly exp-concave losses, that is, when there exists $\alpha > 0$ such that $\exp(-\alpha\ell_t)$ is concave for all $t$. These methods, Exponentially Weighted Online Optimization (EWOO) and Follow The Approximate Leader (FTAL), are summarized in Algorithm 2 and 3 (their Online Newton Step (ONS) algorithm is very similar to FTAL and and therefore omitted). The respective regret bounds are given in Theorems 4 and 7 in (Hazan et al., 2007) and are summarized in Table 1.

**Algorithm 1** Greedy Projection method (GP) a.k.a. Online Gradient Descent (OGD), with input sequence $(\ell_t)$ and learning rates $(\eta_t)$

1: **for** $t \in \mathbb{N}$ **do**
2:  Let $\tilde{s}_{t+1} = s_t - \eta_{t+1} \nabla \ell_t(s_t)$
3:  Update: $x_{t+1} = \delta_{s_{t+1}}$, where

$$s_{t+1} = \arg\min_{s \in S} \|s - \tilde{s}_{t+1}\|$$

---

**Algorithm 2** Exponentially Weighted Online Optimization method (EWOO), with input sequence $(\ell_t)$ and learning rate $\alpha$.

1: **for** $t \in \mathbb{N}$ **do**
2:  Let $L_t = \sum_{\tau=1}^{t} \ell_\tau$
3:  Let $\tilde{x}_{t+1}(s) = \dfrac{e^{-\alpha L_t(s)}}{\int_S e^{-\alpha L_t(s)} \lambda(ds)}$
4:  Update: $x_{t+1} = \delta_{s_{t+1}}$, where

$$s_{t+1} = \mathbb{E}_{s \sim \tilde{x}_{t+1}}[s]$$

---

**Algorithm 3** Follow The Approximate Leader (FTAL) with input sequence $(\ell_t)$ and parameter $\beta$.

1: **for** $t \in \mathbb{N}$ **do**
2:  Let $g_\tau = \nabla \ell_\tau(s_\tau)$
3:  Let $A_t = \sum_{\tau=1}^{t} g_\tau(g_\tau)^T$ and
$\tilde{s}_{t+1} = (A_t)^\dagger \left( \sum_{\tau=1}^{t} g_\tau(g_\tau)^T s_\tau - \frac{1}{\beta} g_\tau \right)$,
and define $\|s\|_{A_t} = \langle s, A_t s \rangle$.
4:  Update: $x_{t+1} = \delta_{s_{t+1}}$, where

$$s_{t+1} = \arg\min_{s \in S} \|s - \tilde{s}_{t+1}\|_{A_t}$$

---

**Algorithm 4** Dual Averaging (DA) with input sequence $(u_t)$, learning rates $(\eta_t)$, and regularizer $h$.

1: **for** $t \in \mathbb{N}$ **do**
2:  Let $U_t = \sum_{\tau=1}^{t} u_\tau$
3:  Update

$$\begin{aligned} x_{(t+1)} &= Dh^*(\eta_t U_t) \\ &= \arg\max_{x \in \mathcal{X}} \langle \eta_t U_t, x \rangle - h(x) \end{aligned}$$

---

**Example S.1** (Convex Quadratics on a Hypercube). *As a first example, we consider quadratic reward functions of the form $u_t(s) = -\frac{1}{2}(s - \mu_t)^T Q_t(s - \mu_t) - c_t$, where $Q_t$ is p.d. symmetric, and $c_t \geq 0$. The domain is $S = \{\|s\|_\infty \leq 0.5\}$ with $D_S = \sqrt{n}$, and the rewards are generated randomly, L-Lipschitz with $\mathrm{L} = 5$ and uniformly bounded by $\|u_t\|_\infty \leq 3.75$ and $\|u_t\|_4 \leq 1.6$. Figure 4 shows the time-average regrets $\mathcal{R}_t/t$ in dimensions $n = 2$ and $n = 3$ for time horizons of $T = 10^4$ and $T = 4 \cdot 10^3$, respectively. Displayed are the empirical means over $N = 2500$ runs of the algorithm (solid), the associated theoretical bounds[1] (dashed), and the regions between the associated 10% and 90% quantiles (shaded).*

*Not surprisingly, those algorithms that exploit the strong convexity of the problem (OGD, FTAL, EWOO) achieve better asymptotic rates than GP (which requires only convexity) or DA (which makes no convexity assumptions at all). Still, the regret of DA is not significantly higher than that of GP and OGD, and is competitive with FTAL over the simulation horizon. We note that the theoretical regret bounds for both DA instances are much closer to the actual regret of the algorithm.*

Figure 4: Time-average regret of different online learning algorithms

*Table 2 shows the decay rates (which correspond to the slopes in the log-log plots) of empirical means and theoretical bounds in Figure 4 at the end of the simulation horizon. There is a relatively good match between bounds and simulations. Except for FTAL and EWOO, all algorithms exhibit a*

*decay that is faster than that of the associated bound[2]. When making this comparison, one must keep in mind that all these bounds are worst-case in nature, and that it is not entirely clear what characterizes a worst-case sequence of reward functions (see Example S.2 for a partial remedy).*

| Algorithm | $n = 2$ | | $n = 3$ | |
|---|---|---|---|---|
| | *simulation* | *theory* | *simulation* | *theory* |
| GP | -0.564 | -0.497 | -0.515 | -0.495 |
| OGD | -0.920 | -0.900 | -0.892 | -0.888 |
| FTAL | -0.780 | -0.900 | -0.705 | -0.888 |
| EWOO | -0.809 | -0.900 | -0.676 | -0.888 |
| DA, Exp | -0.519 | -0.446 | -0.481 | -0.439 |
| DA, 1.5-Norm | -0.452 | -0.333 | -0.396 | -0.286 |

Table 2: Rates in Figure 4

| Potential | simulation | theory |
|---|---|---|
| ExpPot | -0.557 | -0.446 |
| 1.01-Norm | -0.546 | -0.495 |
| 1.05-Norm | -0.477 | -0.476 |
| 1.5-Norm | -0.307 | -0.333 |
| 1.75-Norm | -0.279 | -0.286 |

Table 3: Rates in Figure 5

**Example S.2** (Alternating Affine Losses on a Hypercube). *In this example we consider a situation in which the greedy algorithm mentioned in Section 3 fails[3], and offer a simulation that illustrates the result of Proposition S.2. We consider a sequence of affine reward functions on $S = \{\|s\|_\infty \leq 0.5\}$ in $\mathbb{R}^2$, alternating in such a way that any maximizer $s_t^\star$ of $U_t$ is in fact a minimizer of $U_{t+1}$. Specifically, we choose $u_t(s) = -\langle a_t, s\rangle - c_t$, where*

$$a_0 = [\mathrm{L}/2, 0], \quad c_0 = \mathrm{L}/4, \quad a_t = [(-1)^t \mathrm{L}, 0], \quad c_t = \mathrm{L}/2$$

*for $t \geq 1$. It is easy to see that in this case the greedy algorithm incurs time-average regret $\mathcal{R}_t/t = \mathrm{L} + o(1)$.*

*Figure 5 shows regrets for the greedy algorithm and DA with Exponential and different $\rho$-Norm potentials. Besides the obvious failure of the greedy algorithm, we observe that for p-Norm potentials performance decreases as $\rho \searrow 1$, which can be explained by Proposition S.2. Nevertheless, DA guarantees sublinear regret for any $\rho > 1$ (with theoretical asymptotic rate approaching $t^{-1/2}$ as $\rho \to 1$), though at the cost of much higher constants in the bound as $\rho \approx 1$. Table 3 shows that empirical and theoretical rates in this instance (which is intuitively hard) are very close, providing further support for the theoretical analysis of DA. Finally, Figure 6 for each potential shows the negative entropy $D_{KL}(x_t\|\lambda)$ of $x_t$. From this we observe that the minimizers $x^\star[\rho]$ are indeed more and more concentrated around their mode as $\rho \searrow 1$.*

Figure 5: Failure of the Greedy Policy

Figure 6: Negative entropy of $x_t$

## S.6 Proofs

### Proof of Theorem 1

*Theorem 1.* Essential Fréchet differentiability, the characterization (5) of the Fréchet gradient in (i) and (ii) follow from Theorem S.1, Lemma S.1, and the definition of uniform essential strong convexity. To prove (6), let $\xi_1, \xi_2 \in X^*$ and let $x_i = Df^*(\xi_i) = \arg\max_{x \in X} \langle \xi_i, x \rangle - f(x)$. Then, by first-order optimality, $\langle z - \xi_i, x - x_i \rangle \geq 0$, $\forall z \in \partial f(x_i), \forall x \in X$. In particular,

$$\langle z_1 - \xi_1, x_2 - x_1 \rangle \geq 0$$
$$\langle z_2 - \xi_2, x_1 - x_2 \rangle \geq 0$$

for all $z_i \in \partial f(x_i)$, $i = 1, 2$. Summing these inequalities we find that

$$\langle \xi_1 - \xi_2, x_1 - x_2 \rangle \geq \langle z_1 - z_2, x_1 - x_2 \rangle$$

By uniform strong convexity, we further have that $f(x) \geq f(x_i) + \langle x - x_i, z_i \rangle + \gamma(\|x - x_i\|)$ for all $x \in X$. In particular,

$$f(x_1) \geq f(x_2) + \langle x_1 - x_2, z_2 \rangle + \gamma(\|x_1 - x_2\|)$$
$$f(x_2) \geq f(x_1) + \langle x_2 - x_1, z_1 \rangle + \gamma(\|x_2 - x_1\|)$$

and summing these inequalities yields

$$\langle z_1 - z_2, x_1 - x_2 \rangle \geq 2\gamma(\|x_1 - x_2\|)$$

On the other hand, $\langle \xi_1 - \xi_2, x_1 - x_2 \rangle \leq \|\xi_1 - \xi_2\|_* \|x_1 - x_2\|$ by definition of the dual norm, so

$$\tilde{\gamma}(\|x_1 - x_2\|) \leq \frac{1}{2}\|\xi_1 - \xi_2\|_*$$

using the definition of $\tilde{\gamma}$. If $\tilde{\gamma}$ is strictly increasing it admits a (strictly increasing) inverse $\tilde{\gamma}^{-1}$. Applying $\tilde{\gamma}^{-1}$ to both sides then yields (6). $\qquad\square$

### Proof of Theorem 2

*Theorem 2.* We consider the continuous-time reward and learning rate processes $u^c$ and $\eta^c$ given by $u_t^c := u_{\lceil t \rceil}$ and $\eta^c(t) := \eta_{\lfloor t \rfloor \vee 1}$, respectively, where $\lceil r \rceil := \inf\{n \in \mathbb{Z} : n \geq r\}$ and $\lfloor r \rfloor = \sup\{n \in \mathbb{Z} : n \leq r\}$ for all $r \in \mathbb{R}$ and $a \vee b = \min(a, b)$. In doing so we follow the ideas of the analysis of Kwon and Mertikopoulos (2014) (our problem is, however, different as our reward vectors are infinite-dimensional). With this

$$x_k = Dh^*\left(\eta_{k-1}\sum_{j=1}^{k-1} u_j\right) = Dh^*\left(\eta^c(k-1)\int_0^{k-1} u_\tau^c\, d\tau\right) = x_{k-1}^c$$

and thus, for $j \geq 1$ and $t \in (j-1, j)$, we have

$$|\langle u_t^c, x_t^c \rangle - \langle u_j, x_j \rangle| = |\langle u_j, x_t^c - x_{j-1}^c \rangle| \leq \|u_j\|_* \|x_t^c - x_{j-1}^c\| \tag{13}$$

by definition of the dual norm. Therefore

$$|\langle u_t^c, x_t^c \rangle - \langle u_j, x_j \rangle| \leq \|u_j\|_* \|Dh^*(y_t^c) - Dh^*(y_{j-1}^c)\| \leq \|u_j\|_* \tilde{\gamma}^{-1}\left(\|y_t^c - y_{j-1}^c\|_*/2\right) \tag{14}$$

where the second inequality follows from Theorem 1. From the definition of $y_t^c$, we have

$$\|y_t^c - y_{j-1}^c\|_* = \left\|\eta^c(j-1)\int_{j-1}^t u_\tau^c\, d\tau\right\|_* = \eta_{j-1}\|u_j\|_*(t - j + 1)$$

and therefore

$$\left|\int_0^k \langle u_\tau^c, x_\tau^c \rangle d\tau - \sum_{j=1}^k \langle u_j, x_j \rangle\right| \leq \sum_{j=1}^k \int_{j-1}^j |\langle u_\tau^c, x_\tau^c \rangle - \langle u_j, x_j \rangle|\, d\tau$$

$$\leq \sum_{j=1}^k \|u_j\|_* \int_{j-1}^j \tilde{\gamma}^{-1}\left(\frac{\eta_{j-1}(t-j+1)}{2}\|u_j\|_*\right) d\tau$$

$$\leq \sum_{j=1}^k \|u_j\|_* \tilde{\gamma}^{-1}\left(\frac{\eta_{j-1}}{2}\|u_j\|_*\right)$$

where the last equality follows since $\tilde{\gamma}^{-1}$ is non-decreasing (a consequence of $\gamma$ being sublinear). Finally, we note that

$$\sum_{j=1}^{k}\langle u_j, x\rangle - \sum_{j=1}^{k}\langle u_j, x_j\rangle = \int_0^k \langle u_\tau^c, x\rangle\, d\tau - \sum_{j=1}^{k}\langle u_j, x_j\rangle$$

$$\leq \left|\int_0^k \langle u_\tau^c, x\rangle\, d\tau - \int_0^k \langle u_\tau^c, x_\tau^c\rangle d\tau\right| + \left|\int_0^k \langle u_\tau^c, x_\tau^c\rangle d\tau - \sum_{j=1}^{k}\langle u_j, x_j\rangle\right|$$

The bound (8) then follows from Theorem S.2 and the above. $\qquad\square$

**Proof of Corollary 2**

*Corollary 2.* It is easy to show that $\tilde{\gamma}^{-1}\left(\frac{\eta_{j-1}}{2}\|u_j\|_*\right) \leq (2C)^{-1/\kappa}\eta_{j-1}^{1/\kappa}\|u_j\|_*^{1/\kappa}$. If $\|u_j\|_* \leq M$ for all $j$, then $R_t(x) \leq \frac{h(x)-\underline{h}}{\eta_t^c} + (2C)^{-1/\kappa}M^{1+1/\kappa}\sum_{\tau=1}^{t}\eta_{\tau-1}^{1/\kappa}$. In particular, if $\eta_t = \eta\, t^{-\beta}$, then (10) follows from the bound $\sum_{\tau=1}^{t}(j-1)^{-\beta/\kappa} \leq \int_0^t v^{-\beta/\kappa}dv = \frac{\kappa}{\kappa-\beta}t^{1-\frac{\beta}{\kappa}}$ $\qquad\square$

**Proof of Theorem 3**

*Theorem 3.* The space $X = L^p(S)$ is uniformly convex (Clarkson, 1936), and thus reflexive (Milman, 1938). Its dual is $X^* = L^q(S,\mu)$ for $q = \frac{p}{p-1}$ and $\langle x,\xi\rangle = \int_S x(s)\xi(s)\,\mu(ds)$ for $x \in X$ and $\xi \in X^*$. Fix $t < \infty$. Then for any $s \in S$ and all $x \in \mathcal{B}(s,\vartheta_t)$

$$\langle U_t, x\rangle = \int_{B(s,\vartheta_t)} U_t(s')x(s')\,d\mu(s') = \int_{B(s,\vartheta_t)}\sum_{\tau=1}^{t}u_\tau(s')x(s')\,d\mu(s')$$

$$\geq \sum_{\tau=1}^{t}\int_{B(s,\vartheta_t)}\big(u_\tau(s) - \chi(\vartheta_t)\big)x(s')\,d\mu(s') = U_t(s) - t\,\chi(\vartheta_t)$$

and therefore

$$\mathcal{R}_t = \sup_{s\in S}U_t(s) - \sum_{\tau=1}^{t}\langle u_\tau, x_\tau\rangle$$

$$\leq \sup_{s\in S}\inf_{x\in\mathcal{B}(s,\vartheta_t)}\langle U_t, x\rangle + t\,\chi(\vartheta_t) - \sum_{\tau=1}^{t}\langle u_\tau, x_\tau\rangle$$

$$= \sup_{s\in S}\inf_{x\in\mathcal{B}(s,\vartheta_t)}R_t(x) + t\,\chi(\vartheta_t)$$

and thus (7) follows from (5) in Theorem S.2. $\qquad\square$

**Proof of Proposition 1**

*Proposition 1.* By convexity of $f$, we have that $h(x) = h_\phi(x) \geq f_\phi\big(\int_S \frac{dx}{d\mu}d\mu(s)\big) = f_\phi(1) = 0$ for all $x \in \mathcal{X}$, and thus $\underline{h} = 0$. Furthermore, choosing $x$ as the uniform Radon-Nikodym density w.r.t. $\mu$ on $B(s,\vartheta_t)$, i.e.,

$$x(s') = \frac{\mathbf{1}_{B(s,\vartheta_t)}(s')}{\mu(B(s,\vartheta_t))}$$

we have that

$$h(x) = \int_{S_i}f_\phi(x(s'))\,\mu(ds') = \int_{B(s,\vartheta_t)}f_\phi\left(\frac{1}{\mu(B(s,\vartheta_t))}\right)\mu(ds')$$

$$\leq \min\big(C_0(\vartheta_t)^Q, \mu(S)\big)f_\phi\left(\frac{1}{\mu(B(s,\vartheta_t))}\right)$$

where we used the assumption of $r_0$-local $Q$-regularity and the fact that $\vartheta_t \leq r_0$. It is easy to see that $f_\phi$ is increasing on $[1,\infty)$. Indeed, $f_\phi'(x) = \phi^{-1}(x)$, and $\phi^{-1}(x)$ is increasing by assumption with $\phi^{-1}(1) \geq 0$. Moreover, since $\mu(S) = 1$ by assumption, we have that $\mu(B(s,\vartheta_t)) \leq 1$ for any $s$, so

$$h(x) \leq \min\big(C_0(\vartheta_t)^Q, \mu(S)\big)f_\phi\big(c_0^{-1}(\vartheta_t)^{-Q}\big)$$

Plugging this into the general bound (11) of Theorem 3 yields (13). $\qquad\square$

**Proof of Corollary 3**

*Corollary 3.* Plugging $\tilde{\gamma}(r) = 2r$, $f_\phi(x) = x\log x$ and $\chi(r) = C_\alpha r^\alpha$ into (13) we find that

$$\frac{\mathcal{R}_t}{t} \leq \frac{C_0}{c_0\, t\, \eta_t}\log\big(c_0^{-1}\vartheta_t^{-Q}\big) + C_\alpha\vartheta_t^\alpha + \frac{M^2}{t}\sum_{\tau=1}^{t}\eta_{\tau-1}$$

Letting $\eta_t = \eta\sqrt{\log t}\, t^{-\beta}$ we have that

$$\sum_{\tau=1}^{t}\eta_{\tau-1} \leq \eta\sqrt{\log t}\sum_{\tau=1}^{t}t^{-\beta} \leq \eta\sqrt{\log t}\sum_{\tau=1}^{t}\int_{\tau-1}^{\tau}z^{-\beta}dz = \eta\sqrt{\log t}\int_{0}^{t}z^{-\beta}dz = \frac{\eta\sqrt{\log t}}{1-\beta}t^{1-\beta}$$

and therefore

$$\frac{\mathcal{R}_t}{t} \leq \frac{C_0}{c_0\eta}\frac{t^{\beta-1}}{\sqrt{\log t}}\log\big(c_0^{-1}\vartheta_t^{-Q}\big) + C_\alpha\vartheta_t^\alpha + \frac{\eta M^2}{1-\beta}\sqrt{\log t}\, t^{-\beta}$$

Choosing $\beta = 1/2$ and $\vartheta_t = \vartheta^{\frac{1}{\alpha}}(\log t)^{\frac{1}{2\alpha}}t^{-\frac{\beta}{\alpha}}$ this becomes, after dropping a $1/\log t$ term,

$$\frac{\mathcal{R}_t}{t} \leq \left(\frac{C_0}{c_0\,\eta}\left(\log(c_0^{-1}\vartheta^{-Q/\alpha}) + \frac{Q}{2\alpha}\right) + C_\alpha\vartheta + 2\eta M^2\right)\sqrt{\frac{\log t}{t}}$$

as $\vartheta_t < r_0$ since $\sqrt{\log t/t} < \vartheta^{-1}r_0^\alpha$. Then choosing $\eta = \frac{1}{M}\sqrt{\frac{C_0 Q}{2c_0}\log(c_0^{-1}\vartheta^{-Q/\alpha}) + \frac{Q}{2\alpha}}$ gives

$$\frac{\mathcal{R}_t}{t} \leq \left(2M\sqrt{\frac{2C_0}{c_0}\left(\log(c_0^{-1}\vartheta^{-Q/\alpha}) + \frac{Q}{2\alpha}\right)} + C_\alpha\vartheta\right)\sqrt{\frac{\log t}{t}}$$

$\square$

**Proof of Theorem 4**

For the proof of Theorem 4 we will make use of a few intermediate results.

**Proposition S.5.** *Suppose Assumption 2 holds. Then*

$$\mathcal{R}_t \;=\; \sup_{x\in\mathcal{X}}R_t(x) \;=\; \sup_{x\in\mathcal{P}}R_t(x) \;=\; \sup_{s\in S}U_t(s) - \sum_{\tau=1}^{t}\langle u_\tau, x_\tau\rangle \tag{15}$$

Denote by $\mathbf{1}_A$ the indicator function of the set $A$, i.e. $\mathbf{1}_A(s) = 1$ if $s \in A$ and $\mathbf{1}_A(s) = 0$ if $s \notin A$. We will also make use of the following Lemma:

**Lemma S.2.** *Let $(S, d)$ be a compact metric space and let $\mu$ be an $r_0$-locally $Q$-regular measure on $S$. For $p \geq 1$ let $\mathcal{X}^p := \{x \in L^p(S,\mu) : x \geq 0 \text{ a.s.}, \|x\|_1 = 1\}$. Suppose further that $f : S \to \mathbb{R}$ is continuous. Then*

$$\sup_{s\in S}f(s) = \sup_{x\in\mathcal{P}}\int_S f(s)\,dx(s) = \sup_{x\in\mathcal{X}^p}\int_S f(s)\,dx(s), \qquad \forall\, p\in[0,\infty] \tag{16}$$

*Lemma S.2.* The first equality follows directly by observing that Borel measures measures include measures with finite support. Clearly $\sup_{x\in\mathcal{P}}\int_S f(s)\,dx(s) \geq \sup_{x\in\mathcal{X}^p}\int_S f(s)\,dx(s)$ since $\mathcal{X}^p \subset \mathcal{P}$ for all $p\in[1,\infty]$. Since $L^p \subset L^q$ for all $q \geq p$ it suffices to show the reverse inequality holds for $p = \infty$. Since $S$ is compact and $f$ is continuous, there exists a maximizer $s^\star$ of $f$ on $S$. Let $\epsilon > 0$. By continuity, there exists $\delta > 0$ such that $|f(s) - f(s')| \leq \epsilon$ whenever $d(s, s') < \delta$. Moreover, by local $Q$-regularity of $\mu$ we have that $\mu(B(s^\star, \delta)) > 0$. Now let $x(s) = \frac{1}{\mu(B(s^\star,\delta))}\mathbf{1}_{B(s^\star,\delta)}(s)$. Clearly $x \in \mathcal{X}^\infty$, and

$$\int_{S_i}f(s)\,dx(s) = \frac{1}{\mu(B(s^\star,\delta))}\int_{B(s^\star,\delta)}f(s)\,d\lambda(s) \geq \frac{1}{\mu(B(s^\star,\delta))}\int_{B(s^\star,\delta)}(f(s^\star) - \epsilon)\,d\lambda(s) = f(s^*) - \epsilon$$

Now let $\epsilon \searrow 0$. $\square$

*Proposition S.5.* Recall that

$$R_t(x) = \sum_{\tau=1}^{t} \langle u_\tau, x \rangle - \sum_{\tau=1}^{t} \langle u_\tau, x_\tau \rangle = \int_S U_t(s)\,dx(s) - \sum_{\tau=1}^{t} \langle u_\tau, x_\tau \rangle$$

Clearly $U_t$ is continuous (in fact, with modulus of continuity $t\,\chi(r)$) on $S$ for any $t < \infty$. The equivalence of the suprema then follows from a direct application of Lemma S.2. $\qquad\square$

*Theorem 4.* Since $S$ is compact there exist $s^a, s^b \in S$ such that $d(s^a, s^b) = D_S$. Let $x^a = \delta_{s^a}$ and $x^b = \delta_{s^b}$, where $\delta_s$ denotes the Dirac measure on $S$ at $s$. Let $w : \mathbb{R} \to \mathbb{R}$ be any function with modulus of continuity $\chi$ such that $\|w(d(\,\cdot\,, s^b))\|_q \leq M$. Define $v : S \to \mathbb{R}$ by $v(s) = w(d(s, s^b))$. Using the triangle inequality it is easy to see that $v$ also has modulus of continuity $\chi$. Now observe that

$$\langle v, x^a - x^b \rangle = v(s^a) - v(s^b) = w(d(s^a, s^b)) = w(D_S)$$

Let $V_1, \ldots, V_2$ a sequence of i.i.d. Rademacher random variables, i.e. $\mathbb{P}(V_i = +1) = \mathbb{P}(V_i = -1) = \frac{1}{2}$, and consider the (random) sequence of reward vectors $(u_\tau)_{\tau=1}^t$ with $u_t = V_t v$. By Proposition S.5 we have that $\mathcal{R}_t = \sup_{x \in \mathcal{P}} R_t(x)$, and thus

$$\mathbb{E}[\mathcal{R}_t] = \mathbb{E}\left[ \sup_{x \in \mathcal{P}} \sum_{\tau=1}^{t} \langle u_\tau, x \rangle - \sum_{\tau=1}^{t} \langle u_\tau, x_\tau \rangle \right] \geq \mathbb{E}\left[ \max_{x \in \{x^a, x^b\}} \sum_{\tau=1}^{t} \langle u_\tau, x \rangle \right] - \mathbb{E}\left[ \sum_{\tau=1}^{t} \langle u_\tau, x_\tau \rangle \right]$$

$$= \mathbb{E}\left[ \max_{x \in \{x^a, x^b\}} \sum_{\tau=1}^{t} V_\tau \langle v, x \rangle \right] - \mathbb{E}\left[ \sum_{\tau=1}^{t} V_\tau \langle v, x_\tau \rangle \right]$$

Observe that the second expectation is zero for any sequence of $(x_\tau)_{\tau=1}^t$ with $x_\tau$ measurable with respect to $\sigma(V_1, \ldots, V_{\tau-1})$, i.e. any online algorithm. Noting that $\max(a, b) = \frac{1}{2}(a + b) + \frac{1}{2}|a - b|$ we thus have that

$$\mathbb{E}[\mathcal{R}_t] \geq \frac{1}{2}\mathbb{E}\left[ \sum_{\tau=1}^{t} V_\tau \langle v, x^a + x^b \rangle \right] + \frac{1}{2}\mathbb{E}\left[ \left| \sum_{\tau=1}^{t} V_\tau \langle v, x^a - x^b \rangle \right| \right]$$

$$= \frac{w(D_S)}{2}\mathbb{E}\left[ \left| \sum_{\tau=1}^{t} V_\tau \right| \right] \geq \frac{w(D_S)}{2\sqrt{2}}\sqrt{t}$$

where the last step follows from an application of Khintchine's inequality (Haagerup, 1981). $\qquad\square$

**Proof of Proposition 2**

**Lemma S.3.** *Let $C \in \mathbb{R}$ and $0 < \beta \leq 1$. The function $v : [0, \infty)$ given by $v(r) = Cr^\beta$ is Hölder continuous with modulus of continuity $\chi(r) = |C|^\beta r^\beta$.*

*Lemma S.3.* Noting that $|x + y|^\beta \leq |x|^\beta + |y|^\beta$ for any $x, y \in \mathbb{R}$ we find with $x = Cr_1 - Cr_2$ and $y = Cr_2$ for any $r_1, r_2 \geq 0$ that $|C|r_1^\beta - |C|r_2^\beta \leq |C|^\beta |r_1 - r_2|^\beta$. Exchanging the roles of $r_1$ and $r_2$ then yields $\left| Cr_1^\beta - Cr_2^\beta \right| \leq |C|^\beta |r_1 - r_2|^\beta$. $\qquad\square$

*Proposition 2.* With $s^a, s^b$ as in the proof of Theorem 4, choose

$$w(r) = \min\left( C_\alpha^{1/\alpha},\ M\|d(\,\cdot\,, s^b)^\alpha\|_q^{-1} \right) r^\alpha$$

Then clearly $\|w(d(\,\cdot\,, s^b))\|_q \leq M$ by construction. Moreover, $w$ has modulus of continuity $\tilde{\chi}(r) \leq C_\alpha r^\alpha$ by Lemma S.3. The result follows from observing that $\|d(\,\cdot\,, s^b)^\alpha\|_q \leq \|D_S^\alpha\|_q = D_S^\alpha$. $\qquad\square$

**Proof of Proposition S.2**

*Proposition S.2.* Fix $t < \infty$ and let $\delta > 0$. Consider $x \in \mathcal{X}$ with $\epsilon := \int_{(B_\delta^*)^c} x(s)\mu(ds) > 0$. and define the function $\kappa : \mathbb{R}^+ \to \mathbb{R}^+$ as $\kappa(u) = \sup_{s \in (B_u^*)^c} U_t(s)$. Clearly, $\kappa$ is decreasing, $\kappa(u) < U^*$ for $u > 0$ by definition of $S^*$, and continuous (by continuity of $U_t$). We then have that $U_t(s) < U^* - \kappa(d(s, S^*))$ for all $s \in S$. Let $0 < \delta' < \chi^{-1}(\frac{\kappa(\delta)}{2\,t})$ such that $\mu(B_{\delta'}^*) > 0$. Such a $\delta'$ always exists by $Q$-regularity of $\mu$. Consider

$$\tilde{x}(s) = x(s)\mathbf{1}_{B_\delta^*}(s) + \frac{\epsilon}{\mu(B_{\delta'}^*)}\mathbf{1}_{B_{\delta'}^*}(s)$$

Clearly, $\tilde{x} \in \mathcal{X}$. Furthermore,

$$
\begin{aligned}
(*) &:= \int_S \eta_t U_t(v)\tilde{x}(v)\mu(dv) - h_i(\tilde{x}) - \int_S \eta_k U_k(v)x(v)\mu(dv) + h_i(x) \\
&= \frac{\epsilon}{\mu(B_{\delta'}^*)}\int_{B_{\delta'}^*}\eta_k U_k(v)\mu(dv) - \int_{(B_\delta^*)^c}\eta_t U_k(v)x(v)\mu(dv) - (h_i(\tilde{x}) - h_i(x)) \\
&\geq \epsilon\,\eta_k(U^* - t\chi(\delta')) - \epsilon\eta_t(U^* - \kappa(\delta)) - (h_i(\tilde{x}) - h_i(x)) \\
&\geq \epsilon\,\eta_t(\kappa(\delta) - t\chi(\delta')) - (h_i(\tilde{x}) - h_i(x)) \\
&> \frac{\epsilon\,\eta_t\,\kappa(\delta)}{2\,t} - (h_i(\tilde{x}) - h_i(x))
\end{aligned}
$$

Now $h_i(\tilde{x}) - h_i(x) \to 0$ as $i \to \infty$ by consistency of $(h_i)_{i \geq 0}$. Hence there exists $j < \infty$ such that $(*) > 0$ and thus $x \neq x_j^*$ for all $i \geq j$. Since $\epsilon$ was arbitrary, this shows that $\int_{(B_\delta^*)^c} x_i^*(s)\,\mu(ds) \to 0$ as $i \to \infty$. $\qquad\square$

**Proof of Corollary S.1**

*Corollary S.1.* Let $f : S \to \mathbb{R}$ be continuous and bounded, say $|f(s)| \leq M$ for all $s \in S$. Let $\epsilon > 0$. Since $S$ is compact, $f$ is uniformly continuous, i.e. $\exists \delta > 0$ such that $|f(s) - f(s^*)| < \epsilon/2$ for all $s \in B_\delta^*$. By Corollary S.2 there exists $j < \infty$ such that $x_i^*((B_\delta^*)^c) < \frac{\epsilon}{4M}$ for all $i > j$. Hence

$$\int_S |f(s) - f(s^*)|x_i^*(s)\lambda(ds) < \epsilon/2 \int_{B_\delta^*} x_i^*(s)\mu(ds) + 2M\int_{(B_\delta^*)^c} x_i^*(s)\mu(ds) < \epsilon$$

for all $i > j$. $\qquad\square$

**Proof of Proposition 3**

*Proposition S.6.* Suppose $\mathcal{G}$ has value $V$ and consider a sequence of plays $(s_t^1)_{t \geq 1}$, $(s_t^2)_{t \geq 1}$ and suppose that player 1 has sublinear realized regret. Then

$$\liminf_{t \to \infty}\frac{1}{t}\sum_{\tau=1}^{t} u(s_\tau^1, s_\tau^2) \geq V \tag{17}$$

*Proposition S.6.* This proof uses similar arguments as Theorem 7.2 in Cesa-Bianchi and Lugosi (2006), with modifications to accommodate our more general setting of functions on metric spaces.

Since player 1 has sublinear (realized) regret, by (18) it suffices to show that

$$\sup_{s^1 \in S_1}\frac{1}{t}\sum_{\tau=1}^{t} u(s^1, s_\tau^2) \geq V.$$

Now clearly $\sup_{s^1 \in S_1} f(s^1) = \sup_{x^1 \in \mathcal{P}_1}\int_{S_i} f(s)\,dx^1(s)$ for any $f$ measurable, thus we may equivalently show that $\sup_{x^1 \in \mathcal{P}_1}\frac{1}{t}\sum_{\tau=1}^{t}\int_{S_1} u(s^1, s_\tau^2)\,dx^1(s^1) \geq V$. Observe that, for all $x^1 \in \mathcal{P}_1$,

$$
\begin{aligned}
\frac{1}{t}\sum_{\tau=1}^{t}\int_{S_1} u(s^1, s_\tau^2)\,dx^1(s^1) &= \int_{S_1}\frac{1}{t}\sum_{\tau=1}^{t} u(s^1, s_\tau^2)\,dx^1(s^1) \\
&= \int_{S_1}\frac{1}{t}\sum_{\tau=1}^{t}\Big(\int_{S_2} u(s^1, s)\,d\delta_{s_\tau^2}(s)\Big)\,dx^1(s^1) \\
&= \bar{u}(x^1, \hat{x}_t^2)
\end{aligned}
$$

where $\hat{x}_t^2(B) := \frac{1}{t}\sum_{\tau=1}^t \mathbf{1}_B(s_\tau^2)$ for any Borel set $B \subset S_2$. Since $\hat{x}_t^2 \in \mathcal{P}_2$ we thus have that

$$\sup_{x^1 \in \mathcal{P}_1} \bar{u}(x^1, \hat{x}_t^2) \geq \inf_{x^2 \in \mathcal{P}_2} \sup_{x^1 \in \mathcal{P}_1} \bar{u}(x^1, x^2) = V$$

□

*Proposition 3.* Using the fact that the payoff of player 2 is the negative of player 1, we have from Proposition S.6 and the fact that the game has a value that

$$\liminf_{t \to \infty} \frac{1}{t}\sum_{\tau=1}^t -u(s_\tau^1, s_\tau^2) \geq -V$$

and thus

$$\limsup_{t \to \infty} \frac{1}{t}\sum_{\tau=1}^t u(s_\tau^1, s_\tau^2) \leq V$$

Combining this with (17) yields that $\lim_{t \to \infty} \frac{1}{t}\sum_{\tau=1}^t u(s_\tau^1, s_\tau^2) = V$. □

**Proof of Theorem 5**

In the proof of the theorem we will use the following Lemma:

**Lemma S.4.** *The functions $g_1(x^2) := \sup_{x^1 \in \mathcal{P}_1} \bar{u}(x^1, x^2)$ and $g_2(x^1) := \inf_{x^2 \in \mathcal{P}_2} \bar{u}(x^1, x^2)$ are continuous with respect to the weak topology.*

*Lemma S.4.* It suffices to show that $g_1^{-1}((-\infty, a))$ and $g^{-1}((b, \infty))$ are open, since the sets of the form $(-\infty, a)$ and $(b, \infty)$ form a subbase for the topology of $\mathbb{R}$. Observe first that $u$ is continuous. Indeed, by Assumption 3, we have for any $s, t \in S_1 \times S_2$ that

$$|u(s^1, s^2) - u(t^1, t^2)| \leq |u(s^1, s^2) - u(s^1, t^2)| + |u(s^1, t^2) - u(t^1, t^2)|$$
$$\leq \chi^2(d_2(s^2, t^2)) + \chi^1(d_1(s^1, t^1))$$

and so for any $\epsilon > 0$ there exists $\delta > 0$ such that $|u(s^1, s^2) - u(t^1, t^2)| < \epsilon$ whenever $(d_1 \times d_2)(s, t) < \delta$. Since $u$ is continuous on the compact set $S_1 \times S_2$ it is bounded, i.e. there exists $M < \infty$ such that $|u(s^1, s^2)| \leq M$ for all $s \in S$. This implies that $\bar{u}(x^1, x^2)$ is $2M$-Lipschitz w.r.t the Lévy-Prokhorov metric on $\mathcal{P}_1 \times \mathcal{P}_2$, hence in particular (jointly) continuous w.r.t. the weak (product) topology. Let $\pi_2 : \mathcal{P}_1 \times \mathcal{P}_2 \to \mathcal{P}_2$ denote the canonical projection onto $\mathcal{P}_2$, which by definition of the product topology is continuous. Together with the continuity of $\bar{u}$ this implies that $g_1^{-1}((b, \infty)) = \pi_2 \circ \bar{u}^{-1}((b, \infty))$ is open. Furthermore, note that $\bar{u}(x^1, x^2) < a, \forall x^1 \in \mathcal{P}_1$ whenever $g_1(x) < a$, and hence for any $x^2 \in \mathcal{P}_2$, the set $(x^1, x^2) \in g_1^{-1}((-\infty, a))$ is open. That is, there exists an open cover of $\mathcal{P}_1 \times \{x^2\}$. Now $\mathcal{P}_1$ is compact in the weak topology, which means we can find a finite subcover $\{U_{x^2}^j\}_{j=1}^{n_{x^2}}$ such that $\bigcap_{j=1}^{n_{x^2}} U_{x^2}^j \supset \mathcal{P}_1 \times \{x_2\}$. Taking the union over all $x^2 \in g^{-1}((-\infty, a))$ we have that $g^{-1}((-\infty, a)) = \bigcup_{x^2 \in g^{-1}((-\infty, a))} \bigcap_{j=1}^{n_{x^2}} U_{x^2}^j$, which is an open set. This shows that $g_1$ is continuous. The argument for showing continuity of $g_2$ is essentially the same. □

*Theorem 5.* Note that both $\mathcal{P}_i$ are metrizable and compact in the weak topology (as each $S_i$ is compact), and hence $\mathcal{P}_1 \times \mathcal{P}_2$ by Tychonoff's theorem. Therefore it suffices to show that with probability 1, the weak limit of any weakly converging subsequence of $(\hat{x}_t)_{t=0}^\infty$ is a Nash equilibrium. Let $(\hat{x}_\theta^1, \hat{x}_\theta^2)_{\theta=1}^\infty$ be such weakly convergent subsequence, and $(z^1, z^2) \in \mathcal{P}_1 \times \mathcal{P}_2$ its weak limit. We will show that whenever a given realization of plays $(s_t^1), (s_t^2)$ has sublinear regret for both players, $(z^1, z^2)$ is a Nash Equilibrium, i.e.,

$$\sup_{x^1 \in \mathcal{P}_1} \bar{u}(x^1, z^2) = V = \inf_{x^2 \in \mathcal{P}_2} \bar{u}(z^1, x^2). \tag{18}$$

Let $g_1(x_2) := \sup_{x^1 \in \mathcal{P}_1} \bar{u}(x^1, x^2)$ and $g_2(x^1) := \inf_{x^2 \in \mathcal{P}_2} \bar{u}(x^1, x^2)$, which by Lemma S.4 are continuous w.r.t. the weak topology. Hence, using that $\hat{x}_\theta^i \rightharpoonup z^i$ for $i = 1, 2$, (18) is equivalent to

$$\lim_{\theta \to \infty} \sup_{x^1 \in \mathcal{P}_1} \bar{u}(x^1, \hat{x}_\theta^2) = V, \tag{19a}$$

$$\lim_{\theta \to \infty} \inf_{x^2 \in \mathcal{P}_2} \bar{u}(\hat{x}_\theta^1, x^2) = V. \tag{19b}$$

We first show (19a). By assumption, the game has value $V$, i.e. it holds that $\inf_{x^2 \in \mathcal{P}_2} \sup_{x^1 \in \mathcal{P}_1} \bar{u}(x^1, x^2) = V$ and thus, in particular, that

$$\liminf_{\theta \to \infty} \sup_{x^1 \in \mathcal{P}_1} \bar{u}(x^1, \hat{x}_\theta^2) \geq V. \tag{20}$$

Now, suppose that for a realization $(s_\tau^1), (s_\tau^2)$, the regret of the second player is sublinear, i.e.

$$\limsup_{t \to \infty} \frac{1}{t} \left( \sup_{x^1 \in \mathcal{P}_1} \sum_{\tau=1}^t \int_{S_1} u(s^1, s_\tau^2) \, dx^1(s^1) - \sum_{\tau=1}^n u(s_\tau^1, s_\tau^2) \right) \leq 0.$$

Then by Proposition 3, $\lim_{t \to \infty} \frac{1}{t} \sum_{\tau=1}^t u(s_\tau^1, s_\tau^2) = V$, and we have

$$V \geq \limsup_{t \to \infty} \sup_{x_1 \in \mathcal{P}_1} \frac{1}{t} \sum_{\tau=1}^t \int_{S_1} u(s^1, s_\tau^2) \, dx^1(s^1)$$

$$= \limsup_{t \to \infty} \sup_{x_1 \in \mathcal{P}_1} \int_{S_1} \frac{1}{t} \sum_{\tau=1}^t u(s_1, s_\tau^2) \, dx^1(s^1)$$

$$= \limsup_{t \to \infty} \sup_{x_1 \in \mathcal{P}_1} \bar{u}(x^1, \hat{x}_t^2)$$

$$\geq \limsup_{\theta \to \infty} \sup_{x_1 \in \mathcal{P}_1} \bar{u}(x^1, \hat{x}_\theta^2).$$

Combining the last inequality with (20) proves (19a). The argument for (19b) is essentially the same, modulo some sign changes.

This proves that for any realization with sublinear regret for both players, all weak limit points of the sequence $(\hat{x}_t^1, \hat{x}_t^2)$ lie in the set of Nash equilibria. But by definition of Hannan consistency, this happens with probability 1.

$\square$

**Proof of Theorem 6**

*Theorem 6.* To start, note that for any $p > 1$ the space $\mathcal{X}$ as a closed subset of $L^p(S, \mu)$ is a complete metric space, hence Polish and thus there exists a Borel isomorphism between $\mathcal{X}$ and the Lebesgue measure on the unit interval. Consequently, to randomize its plays according to a sequence of probability measures in $\mathcal{X}$, it suffices that player $i$ has access to a sequence of i.i.d. random variables drawn from the uniform distribution on $[0, 1]$. Denote this sequence by $Z^i = (Z_1^i, Z_2^i, \dots)$.

The key observation is that if player $-i$ plays a non-oblivious strategy, then the partial rewards will not be some a priori fixed sequence of reward functions, but will depend on the history of play. Indeed, since $\tilde{u}_t^i(\,\cdot\,) = \sum_{\tau=1}^t u_i(\,\cdot\,, s_\tau^{-i})$ and since $s_\tau^{-i}$ is itself some function of past plays $s_1^i, \dots, s_{\tau-1}^i$, the partial reward functions $\tilde{u}_t^i$ are measurable w.r.t. the $\sigma$ field generated by $(Z_1^i, \dots, Z_t^i)$. Note that this implicitly assumes that any randomization performed by player $-i$ is independent of that of player $i$. Let $\mathbb{E}_t^i[X] := \mathbb{E}[X \mid Z_1^i, \dots, Z_{t-1}^i]$ denote the conditional expectation of $X$ given the past plays of player $i$. Then

$$\sum_{\tau=1}^t u^i(s^i, s_\tau^{-i}) - \sum_{\tau=1}^t \mathbb{E}_\tau^i[u^i(s_\tau^i, s_\tau^{-i})] \leq \sum_{\tau=1}^t \sup_{s_\tau^{-i}} \mathbb{E}_\tau^i[u_i(s^i, s_\tau^{-i}) - u_i(s_\tau^i, s_\tau^{-i})]$$

$$= \sum_{\tau=1}^t \sup_{\tilde{u}_\tau^i} \mathbb{E}_\tau^i[\tilde{u}_\tau^i(s^i) - \tilde{u}_\tau^i(s_\tau^i)]$$

$$= \sup_{\tilde{u}_1^i, \dots, \tilde{u}_t^i} \sum_{\tau=1}^t \mathbb{E}_\tau^i[\tilde{u}_\tau^i(s^i) - \tilde{u}_\tau^i(s_\tau^i)] \tag{21}$$

where the last step uses the fact that $s_\tau^i \sim x_\tau^i := Dh_i^*\big(\eta_{\tau-1}\sum_{\theta=1}^{t-1}\tilde{u}_\theta^i\big)$, which depends on the sequence $\{s_\theta^i\}_{\theta=1}^{\tau-1}$ only through the sequence $\{\tilde{u}_\theta^i\}_{\theta=1}^{\tau-1}$ of observed partial loss functions.

From Proposition S.5 we have that

$$\mathcal{R}_t = \sup_{s^i \in S_i} \sup_{\tilde{u}_1^i,\ldots,\tilde{u}_t^i} \sum_{\tau=1}^t \tilde{u}_\tau^i(s^i) - \sum_{\tau=1}^t \langle \tilde{u}_i^\tau, x_\tau^i \rangle = \sup_{s^i \in S_i} \sup_{\tilde{u}_1^i,\ldots,\tilde{u}_t^i} \sum_{\tau=1}^t \mathbb{E}_\tau^i\big[\tilde{u}_\tau^i(s^i) - \tilde{u}_\tau^i(s_\tau^i)\big] \quad (22)$$

Now let $W_\tau^i = \tilde{u}_\tau^i(s_\tau^i) - \langle \tilde{u}_\tau^i, x_\tau^i \rangle$ and observe that $W_\tau^i$ is a martingale. Indeed,

$$\mathbb{E}[W_\tau^i \mid W_\tau^i,\ldots,W_i^{\tau-1}] = \mathbb{E}[W_\tau^i \mid Z_i^\tau,\ldots,Z_i^{\tau-1}] = 0 \qquad \text{a.s.}$$

Moreover, since by assumption $u_i$ is continuous on the compact set $S_1 \times S_2$, we have that $u_i$ is bounded and therefore $|W_\tau^i - W_{\tau-1}^i| \le M$ for some $M < \infty$. Noting that $W_\tau^i = 0$ it follows from the Azuma-Hoeffding inequality that, for every $\epsilon > 0$, $\mathbb{P}(W_\tau^i \le \epsilon) \ge 1 - \exp(-\frac{\epsilon^2}{2\tau M^2})$ and thus

$$\mathbb{P}\big(\textstyle\sum_{\tau=1}^t W_\tau^i \le M\sqrt{2t\log(t/\epsilon)}\big) \ge 1 - \epsilon \qquad \forall \epsilon > 0$$

Now $\sum_{\tau=1}^t W_\tau^i = \sum_{\tau=1}^t \tilde{u}_\tau^i(s_\tau^i) - \sum_{\tau=1}^t \langle \tilde{u}_\tau^i, x_\tau^i \rangle$, and hence, using (22) and (21), we have for all $t < \infty$ that

$$\sup_{s^i \in S_i} \frac{1}{t}\bigg(\sum_{\tau=1}^t u_i(s^i, s_\tau^{-i}) - \sum_{\tau=1}^t u_i(s_\tau^i, s_\tau^{-i})\bigg) \le \frac{\mathcal{R}_t}{t} + M\sqrt{\frac{2\log(t/\epsilon)}{t}}$$

Now $\mathcal{R}_t/t \to 0$ by assumption, and $\sqrt{\frac{\log(t/\epsilon)}{t}} \to 0$ for any $\epsilon > 0$, which proves Hannan consistency. $\qquad\square$

## Footnotes

[1]For easier readability we omitted the bound on FTAL, which in this example is much higher than the others.

[2]For EWOO this discrepancy is likely due to numerical inaccuracies at the very small regrets for large $t$, while for FTAL the simulation may not have reached the asymptotic regime yet.

[3]In fact, any deterministic policy will incur linear regret in a nontrivial adversarial setting.