[Reviews · NeurIPS 2016]

Reviewer 1

Summary

The paper addresses an adversarial online learning problem on a reflexive Banach space. The authors generalize the method of Dual Averaging to the setting and obtain upper bounds on the worst-case regret using results from infinite dimensional convex analysis. Making no convexity assumptions on either S or the reward functions, the author also obtains explicit regret bounds in a setting where the decision set is the set of probability distributions on a compact metric space S. The author also proves a general lower bound on the worst-case regret for any online algorithm and apply these results to the problem of learning in repeated two-player zero-sum games on compact metric spaces. Besides, the authors claims that they first prove that if both players play a Hannan-consistent strategy, then with probability 1 the empirical distributions of play weakly converge to the set of Nash equilibria of the game. Under mild assumptions, Dual Averaging on the (infinite-dimensional) space of probability distributions indeed achieves Hannan-consistency.

Qualitative Assessment

The main contribution of this paper is that the regret bound of a general adversarial online learning problem is extended from the finite sets to infinite sets, and the author solves several issues raised in the paper. However, there is no conclusion part in this paper. The structure of this paper is too tight, and there is partial redundant. For example, I think section 4.1 can be deleted. Hence, the appropriate adjustments and deletions are necessary. I doubt the correctness of the second conclusion, i.e, equation (10), in corollary 2 because I found a serious mistake in the proof. The focus of paper should be to explain the problems you have solved and the results you have obtained clearly, rather than simply list the conclusions of the literatures. The paper may be fitter for COLT than NIPS. Minor issues: Line 78 I do not understand why lim sup_t⁡[R_t/t] <= 0. Line 101 I think f > -∞ is unreasonable. Line 260 I do not know why formula (18) satisfies <= 0. Line 304 I cannot understand the formula (19). Line 306 The same problem with lim sup_t⁡[R_t/t] <= 0. Line 311 u(s^1,s^2 )=s^2 s^2-a^1 s^1-a^2 s^2 may be u(s^1,s^2 )=s^1 s^2-a^1 s^1-a^2 s^2. Line 313 U_t^2 (s^2 )=(a^2 t-∑_(τ=1)^ts_τ^1 ) s_2-a^1 ∑_(τ=1)^ts_τ^1 may be U_t^2 (s^2 )=(a^2 t-∑_(τ=1)^ts_τ^1 ) s^2-a^1 ∑_(τ=1)^ts_τ^1 . Line 316 α_t^2=η_t (a^1 t-∑_(τ=1)^ts_τ^1 ) may be α_t^2=η_t (a^2 t-∑_(τ=1)^ts_τ^1 ). I hope the authors can make a detailed explanation of the above issues and revise these errors.

Confidence in this Review

2-Confident (read it all; understood it all reasonably well)


Reviewer 2

Summary

The paper considers the problem of regret minimization in reflexive Banach spaces, and the main contribution is showing that properly tuned variants of dual averaging/FTRL can guarantee sublinear regret bounds in this setting. This result is then applied to the special case of regret minimization in compact metric spaces, where the authors are able to show explicit near-optimal rates that hold uniformly for all comparators. Finally, the authors apply their techniques for proposing algorithms for finding epsilon-Nash equilibria in continuous zero-sum games. The results are illustrated with several simple experiments and examples.

Qualitative Assessment

POST-REBUTTAL COMMENTS Thanks for the replies. I've found the example of X=L2(R) and \cal{X} = \Delta(X) to be convincing enough, so I understand now how the results are more general than those of Sridharan and Tewari (2010). ============================= The paper is very technical and offers a rather demanding reading experience: the main paper is essentially a densely compressed collection of definitions and theorems for a number of different problem settings. The paper is made complete by a tremendous 17-page appendix that not only includes the full proofs, but adds several simulations, illustrative examples and technical results that are interesting in their own right. This structure itself is my main concern about the paper: I am not sure that NIPS is a suitable venue for such enormous papers with so much to say. Splitting papers so that the main file includes the main result and the supplement presents the proofs is considered normal by now; this current paper goes way beyond. Despite my above comments, if the reader is familiar with the underlying concepts, the logic of the paper is reasonably easy to follow. The proofs are clean and appear to be correct. The techniques themselves are not very original: notably, the proof of the key result (Theorem 3) closely follows the analysis of Kwon and Mertikopoulos (2014). Other proof techniques are also familiar from related work. That said, I do appreciate the clarity and the generality of the new results in the current paper. There is room for improvement in discussing the related literature. In particular, the authors write off the result of Sridharan and Tewari (COLT 2010) as being weaker than the new results because their bounds require a uniformly bounded strategy set. However, the results in the current paper also seem to make a similar assumption (Assumption S.2 in the appendix that seems to be necessary for proving all the results in the paper!), so I wouldn't consider this to be a serious distinction. Also, the authors fail to mention the follow-up work by Srebro, Sridharan and Tewari (NIPS 2011) who prove even more general results. I realize that some of the results in that work are way beyond the scope of this paper, but still, a short discussion would be useful to have. A discussion about how the bound of Corollary 3 relates to other results about continuous exponential weights would be even more useful. Overall, while I'm not sure that the structure of the paper is really suitable for NIPS, I think that this is a solid technical paper that may warrant some interest. Detailed comments ================= 020: Is Audibert and Bubeck (2009) really the best reference to illustrate this point? Maybe Cesa-Bianchi and Lugosi (2006) and Shalev-Shwartz (2011) are better references. 049: "existence" -> "existence of" 076: This is a strange way to define worst-case regret; worst-case is usually defined by taking the supremum over the sequence of losses, not the comparator. 086: "w.r.t. to" -> "w.r.t."; this occurs several times in the paper. 110: "strictly convex" was not defined formally, the notation A^{-1} for sets A likewise. Perhaps it would be good to give a reference to a convex analysis textbook for interested readers. 140: This theorem seems to require Assumption S.2 to hold. If so, please state this assumption in the main text. 149: Regret bounds that depend on the comparator x are sometimes the best one can hope for; see, e.g., McMahan and Orabona (COLT 2014). In particular, for the case of unbounded decision sets, it may not be reasonable to expect a bound that holds simultaneously for all x. 222: Also note that this choice of regularizer coincides with the entropic barrier of Bubeck and Eldan (COLT 2015) for Euclidean spaces. 237: "w is any function with modulus of continuity"---does the sequence depend on w? If so, please make it clear in the statement of the theorem. Appendix: 254, Eq. (13): Inconsistent notation for inner products in the first term. 256, display: shouldn't the inequality be an equality instead? 268, display: what is "d\tau"?

Confidence in this Review

3-Expert (read the paper in detail, know the area, quite certain of my opinion)


Reviewer 3

Summary

This paper generalizes several results in online learning to an infinite-action setting, using the formalism of reflexive Banach spaces. The authors provide a generalization of Follow the Regularized Leader (FTRL) and show that it continues to have no regret, and give lower bounds which hold for any online algorithm. A key technical ingredient, of interest in its own right, is a generalization of strong convexity/smoothness duality to reflexive Banach spaces.

Qualitative Assessment

I liked the paper: the results are comprehensive and constitute a solid extension of online learning to infinite action spaces. In the context of NIPS, however, this may be the most abstract paper I have encountered, and close to it even at COLT. To really be appropriate for NIPS, I would have expected the authors to give some substantial motivation for why this generalization is interesting and important for the machine learning community. Are there infinite-action cases of particular interest where this sheds new light? I.e., what are applications of this theory which justify the generalization? The examples given are useful, and a step in the right direction, but even more concrete examples ("toy" or otherwise) would be ideal. I would ask the authors to please comment on the applications/motivation in the response.

Confidence in this Review

1-Less confident (might not have understood significant parts)


Reviewer 4

Summary

The main result of the authors is that - in a continuous two-player zero sum game over compact (not necessarily convex) metric spaces, if both players follow a Hannan-consistent strategy, then with probability 1 their empirical distributions of play weakly converge to the set of Nash equilibria of the game.

Qualitative Assessment

My main concern with the paper is that it is extremely difficult to parse. It would help if the authors provided some intuition as to why some results might hold or not. An important example of continuous 2-player games on convex compact sets are those where the decision sets are combinatorial polytopes - so any point in the polytope is a feasible strategy. This would be an easy link to some settings that are well-studied in the literature, but I don't see any mention of it. It is interesting that they can extend the results to non-convex decision sets, but some intuition why this hold would be nice. Further, the authors remark that the game is full-information - how is then the payoff function for the infinite decision set given? This inevitably imposes a certain structure on the payoff functions if one needs the algorithm to be efficient in any sense - perhaps linear? This I think is an important point to be clarified in the paper.

Confidence in this Review

1-Less confident (might not have understood significant parts)


Reviewer 5

Summary

Presents new bounds for regret minimization in a class of games with infinite action sets (two-player zero-sum games over compact metric spaces). This appears to be first algorithm/bound for this setting. The evaluation is primarily theoretical, and there is an empirical example to provide some intuition.

Qualitative Assessment

The paper is novel, appears to be technically sound, and is clear. It could use better discussion for motivation, and it is not clear to me what, if any, potential impact would be. I am looking at the Cesa-Bianchi book for the result for marginal distributions of Hannan-consistent strategies converging to the set of Nash equilibria. What chapter/page is this? I am not understanding that result. What does it mean for a marginal distribution to converge to the set of Nash equilibria? Each marginal distribution would only converge to one mixed strategy (or not at all). Does this mean that for every Nash equilibrium, there exists some Hannan-consistent strategy for the players that converges to it? I realize that regret minimization is a well studied goal, and that it is new and interesting to explore it in continuous games. But there is no justification for why we care about it other than the citations that prior works have studied it. Is the goal to construct learning/opponent exploitation algorithms for playing against suboptimal opponents? Or is it to compute an approximate equilibrium in self play? What, if any, learning/exploitation and equilibrium computation algorithms are known for this setting? How would this algorithm compare to those? Are there interesting continuous games that motivate this algorithm or that we would now be able to solve that we couldn't before? Or is the goal of the paper just to be a theoretical contribution? The paper end abruptly with a theorem and example, and no concluding section to describe these bigger picture questions and future trajectories. Are the results here surprising? They say that this the first sublinear regret algorithm for continuous games. Was it expected that this would exist for these games? The examples are helpful for understanding the problem. I think it would also be helpful to discuss at least one motivating real-world problem for which the approach could potentially be useful (if an eventual goal is to have practical impact), or make it clear that the goal is just theoretical. What is the gist of the Glicksberg existence result for Nash equilibria in continuous games?

Confidence in this Review

2-Confident (read it all; understood it all reasonably well)


Reviewer 6

Summary

This paper analyzes regret minimization in reflexive Banach spaces, where the loss sequence may be adversarial. The authors show that the DA (Dual Averaging) procedure ensures a low regret in case that the loss functions are linear (under several assumptions). The authors also provide lower bounds, and a result regarding continuous zero sum games, where they show that any regret minimization strategy is Hannan consistent. They also show that in case both players are using regret minimization strategy then we may calculate an approximate Nash Equlibria.

Qualitative Assessment

The contribution of this work is very incremental. Specifically, results similar to Theorems 2 and 3, as well as the subsequent propositions are already well known for DA. The type of extensions to zero sum games is also quite well known. As well as the idea of "linearizing" non-convex problems by extending the decision set to be the set of all distributions over the original set. I therefore do not think it should be accepted to NIPS.

Confidence in this Review

2-Confident (read it all; understood it all reasonably well)